# Turning Stale Gradients into Stable Gradients:
# Coherent Coordinate Descent with Implicit Landscape Smoothing for Lightweight Zeroth-Order Optimization

**Chen Liang** [1]  **Xiatao Sun** [1]  **Qian Wang** [1]  **Daniel Rakita** [1]

## Abstract

Zeroth-Order (ZO) optimization is pivotal for scenarios where backpropagation is unavailable, such as memory-constrained on-device learning and black-box optimization. However, existing methods face a stark trade-off: they are either sample-inefficient (e.g., standard finite differences) or suffer from high variance due to randomized estimation (e.g., random subspace methods). In this work, we propose Coherent Coordinate Descent (CoCD), a deterministic, sample-efficient, and budget-aware ZO optimizer. Theoretically, we formalize the notion of gradient coherence and demonstrate that CoCD is equivalent to Block Cyclic Coordinate Descent (BCCD) with "warm starts," effectively converting historical (stale) gradients from a liability into a computational asset. This mechanism enables $O(1)$ query complexity per step while maintaining global descent directions. Furthermore, we derive error bounds revealing a counter-intuitive insight: larger finite-difference step sizes can induce an implicit smoothing effect on the optimization landscape by reducing the effective smoothness constant, thereby improving convergence stability. Experiments on MLP, CNN, and ResNet architectures (up to 270k parameters) demonstrate that CoCD significantly outperforms BCCD in terms of sample efficiency and convergence loss/accuracy, and exhibits superior stability over randomized ZO methods. Our results suggest that deterministic, structure-aware updates offer a superior alternative to randomization for lightweight ZO optimization.

---

[1]Department of Computer Science, Yale University, New Haven, USA. Correspondence to: Chen Liang, Xiatao Sun, Qian Wang, and Daniel Rakita <{dylan.liang, xiatao.sun, peter.wang.qw262, daniel.rakita}@yale.edu>.

*Proceedings of the 43rd International Conference on Machine Learning*, Seoul, South Korea. PMLR 306, 2026. Copyright 2026 by the author(s).

## 1. Introduction

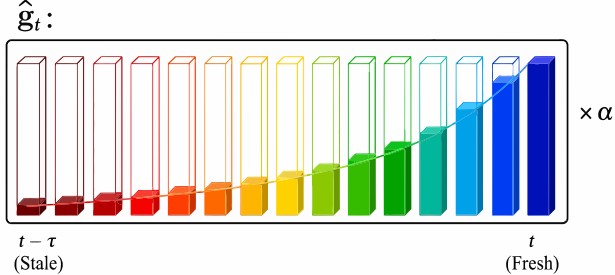

*Figure 1.* Conceptual illustration of CoCD. Each cuboid represents a gradient estimate projected onto a block of coordinates, from the stalest one evaluated at $t - \tau$ to the freshest at $t$. The fill levels encode how the momentum term ($\gamma$) in CoCD assigns exponentially higher weights to more recent gradient estimates. These decay-weighted gradient components are accumulated into $\hat{\mathbf{g}}_t$ and scaled by the learning rate $\alpha$ to form the update direction, illustrating how CoCD leverages gradient history for stable optimization.

Zeroth-Order (ZO) optimization (Liu et al., 2020; Wang et al., 2018), also known as derivative-free optimization (Golovin et al., 2020), serves as a cornerstone for machine learning applications where gradient information is either inaccessible or computationally prohibitive. Key scenarios include black-box adversarial attacks (Ilyas et al., 2018), simulation-based reinforcement learning (Kim et al., 2021), and training on memory-constrained edge devices (Lin et al., 2022; Cai et al., 2020) where constructing a computation graph for backpropagation is infeasible.

Despite its broad applicability, scaling ZO optimization to high-dimensional problems remains a fundamental challenge due to the sample efficiency–variance trade-off. Classical approaches relying on coordinate-wise finite differences (FD) (Thomas, 2013) provide low-variance gradient estimates but scale poorly, requiring $O(d)$ function evaluations per gradient step for $d$ parameters. Conversely, modern randomized approaches, such as Evolution Strategies (ES) (Hansen et al., 2015), Simultaneous Perturbation Stochastic Approximation (SPSA) (Maryak & Chin, 2001), or random subspace methods (e.g., DeepZero (Chen et al., 2024)), reduce the per-step query cost but introduce significant variance. This stochasticity often necessitates small

learning rates or large batch sizes to dampen noise, hindering the wall-clock convergence speed they aim to accelerate.

In this work, we propose a deterministic, cyclic-style coordinate optimization framework (Canutescu & Dunbrack Jr, 2003) that challenges two prevailing assumptions in the zeroth-order optimization literature. The first assumption is that stale gradients, i.e., derivatives computed at previous iterations, are inherently harmful "lags" that should be discarded. We argue instead that optimization trajectories exhibit temporal coherence: gradients evolve continuously over time, with changes bounded by the function's Lipschitz smoothness. Under this perspective, stale gradients can be repurposed as cost-free approximations of the current geometry, effectively providing a "warm start" for the optimizer. The second assumption is that the finite difference interval ($\epsilon$) must be minimized to closely approximate the true gradient. Counterintuitively, we show that larger step sizes can be advantageous. Rather than merely degrading gradient accuracy, a larger $\epsilon$ can estimate the gradient of a smoothed function, implicitly filtering out high-frequency irregularities in the landscape and reducing the effective smoothness constant, thereby enabling more stable descent.

To operationalize these insights, we introduce *Coherent Coordinate Descent* (CoCD). Algorithmically, CoCD cyclically traverses coordinates while maintaining a First-In-First-Out (FIFO) buffer of past gradient estimates. This structure provides explicit control over both the compute budget (active queries per iteration) and the memory budget (stored gradient history), while avoiding the high variance inherent to randomized zeroth-order estimators.

Theoretically, we connect this seemingly heuristic strategy to established optimization theory by proving that CoCD is equivalent to Block Cyclic Coordinate Descent (BCCD) (Beck & Tetruashvili, 2013) equipped with stale-gradient warm starts. This equivalence enables a convergence analysis that explicitly quantifies the approximation error induced by reusing outdated information. Crucially, the resulting bound is governed by the smoothness of the objective. We show that larger finite difference intervals ($\epsilon$) can implicitly smooth the landscape by reducing the effective smoothness constant ($L_\epsilon$), thereby permitting larger step sizes and longer gradient histories without sacrificing stability. Together, these effects translate into faster practical convergence.

Our main contributions are summarized as follows:

- **Algorithmic framework:** We introduce CoCD, a deterministic and memory-efficient zeroth-order optimizer that replaces randomized gradient estimation with structured cyclic updates stored in a FIFO buffer, achieving low query complexity without incurring high variance.

- **Theoretical grounding:** We prove that CoCD is equivalent to BCCD with warm starts and derive convergence guarantees that formally justify gradient reuse and the smoothing effect of larger finite difference steps. Under mild assumptions, CoCD achieves linear convergence in the worst case.

- **Empirical evaluation:** We evaluate CoCD on neural network training tasks, including MLPs for regression, as well as CNNs and ResNet-20 (He et al., 2016) for classification, in the small-to-medium parameter regime (up to 270k). CoCD consistently outperforms BCCD in both sample efficiency and final accuracy, and exhibits superior stability over randomized zeroth-order methods such as SPSA.

We release an open-source implementation of CoCD to support reproducibility and further research[1].

## 2. Related Works

This section situates CoCD within three closely related lines of research: zeroth-order optimization, coordinate descent methods, and optimization with stale gradients.

### 2.1. Zeroth-Order Optimization

Zeroth-order (ZO) methods allow for optimization without explicit gradient computation. While early approaches relied on standard finite differences, recent works have focused on scalability via randomization. Evolution Strategies (Hansen et al., 2015) and Gaussian smoothing (Gao & Sener, 2022) utilize Randomized Gradient Estimation (RGE). However, RGE suffers from high variance, requiring large batch sizes to reduce approximation error.

More recently, DeepZero (Chen et al., 2024) scaled ZO to deep neural networks by combining random subspace optimization with cyclic coordinate updates. Empirical evidence and recent theoretical advances (Gurbuzbalaban et al., 2017; Cai et al., 2023) suggest that Cyclic Gradient Estimation (CGE) can yield significantly better convergence behavior than RGE when navigating non-convex optimization landscapes. However, to accommodate million-scale models, DeepZero effectively compromises on this insight: they apply CGE only after restricting the search to a randomly selected subspace of parameters using "pruning at initialization" techniques (Wang et al., 2020). This approach introduces a structural bias: parameters outside the random subspace are effectively frozen during iterations, potentially missing critical descent directions.

In contrast, CoCD targets memory-constrained regimes where such compromise is unnecessary. We implement true

---

[1] https://github.com/chen-dylan-liang/CoCD

cyclic coordinate descent over the entire parameter space, fully exploiting the theoretical superiority of CGE without the blind spots introduced by random subspace pruning.

## 2.2. Coordinate Descent

Coordinate descent (CD) methods (Wright, 2015) iteratively update a subset of parameters while keeping others fixed. The theoretical analysis of CD has historically bifurcated into randomized (RCD) and cyclic (CCD) approaches. RCD is generally easier to analyze because the expected update direction aligns with the true gradient (Lu & Xiao, 2015). In contrast, analyzing CCD is significantly more challenging due to the deterministic nature of the updates, requiring nontrivial techniques to bound the error propagation across cycles.

Seminal works have established the non-asymptotic (Saha & Tewari, 2013) and finite-time convergence (Saha & Tewari, 2010) properties of plain CCD where gradient estimates are updated one coordinate per iteration. Most recently, Cai et al. (2023) provided a non-asymptotic convergence guarantee for BCCD working on non-convex optimization where multiple coordinates are updated per iteration. However, this rich body of literature operates almost exclusively within the *first-order* context, assuming access to exact partial derivatives. In our work we extend these CCD-based optimization methods to the *zeroth-order* setting, showing that the stability can be maintained even when gradients are approximated via finite differences and contain "stale" information.

## 2.3. Optimization with Stale Gradients

In distributed and asynchronous optimization (Dutta et al., 2018; Zhou et al., 2022), stale gradients typically arise as a byproduct of system latency and are treated as noise to be bounded. Diverging from this perspective, the recent work Web of Affine Spaces Optimization (WASP) (Rakita et al., 2025; Liang & Rakita, 2025) explicitly assumes temporal coherence in optimization trajectories and treats past gradients as informative geometric constraints rather than detrimental artifacts.

WASP operationalizes this idea through structured affine subspace constructions and associated matrix computations, making it well suited for robotics and control applications where objective functions are highly structured but typically involve only a few hundred variables. In contrast, our CoCD framework is designed for significantly lighter-weight operation. By relying almost exclusively on vector-level computations and simple gradient buffering, CoCD avoids expensive matrix factorizations and scales naturally to machine-learning–based objectives with many thousands of parameters. While conceptually inspired by WASP's view of gradient coherence, CoCD targets a distinct computational regime, enabling practical gradient reuse in large-scale zeroth-order optimization where heavier geometric constructions would be prohibitive.

## 3. Preliminaries

In this section, we introduce the problem formulation, notation, and optimizer state representation used throughout the paper.

### 3.1. Problem Formulation and Notation

We consider the zeroth-order optimization problem of minimizing a differentiable objective function $f : \mathbb{R}^n \to \mathbb{R}$ where explicit gradient information is unavailable. The goal is to find

$$\mathbf{x}^* = \arg \min_{\mathbf{x}} f(\mathbf{x})$$

relying solely on function evaluations. We denote the standard basis vectors of $\mathbb{R}^n$ as $\{\mathbf{e}_1, \ldots, \mathbf{e}_n\}$, where $\mathbf{e}_i$ has a 1 in the $i$-th coordinate and 0 elsewhere.

We denote the sequence of decision variables generated by the optimizer as $\mathbf{x}_1, \mathbf{x}_2, \ldots, \mathbf{x}_t$, and the corresponding function values as $f_t \triangleq f(\mathbf{x}_t)$. We use $\mathbf{g}_t \triangleq \nabla f(\mathbf{x}_t)$ to denote the true gradient at step $t$. Since $\mathbf{g}_t$ is inaccessible, the optimizer maintains an estimate of the gradient, denoted as $\hat{\mathbf{g}}$, which is used directly as the descent direction.

### 3.2. Optimizer State and Cyclic Dynamics

A key innovation of our framework is the decoupling of the *optimization step* index $t$ from the *gradient estimation state*. To formalize this, we define the *optimizer state* at any given moment as the tuple

$$\mathcal{S}^{(k)} = \left( \hat{\mathbf{g}}^{(k)}, i^{(k)} \right), \tag{1}$$

where:

- $\hat{\mathbf{g}}^{(k)} \in \mathbb{R}^n$ represents the current global gradient estimate maintained by the optimizer.

- $i^{(k)} \in \{1, \ldots, n\}$ denotes the *active coordinate index* scheduled for the next finite difference query.

- The superscript $(k)$ indexes internal state updates and is distinct from the outer optimization iteration counter $t$.

For a single optimization step at $\mathbf{x}_t$, the optimizer may perform a fixed number of function queries to refine its gradient estimate. We refer to this number as the *compute budget*, denoted by $B$. Consequently, for each position $\mathbf{x}_t$, the optimizer state undergoes $B$ internal updates.

The state transition follows a deterministic, cyclic structure. The active coordinate index updates according to

$$i^{(k+1)} = (i^{(k)} \bmod n) + 1. \tag{2}$$

Simultaneously, the gradient estimate $\hat{\mathbf{g}}^{(k)}$ is updated by replacing the entry corresponding to the active coordinate with the most recent finite difference information. Let $\tilde{\nabla}_i f(\mathbf{x})$ denote the coordinate-wise finite difference approximation (e.g., central difference):

$$\tilde{\nabla}_i f(\mathbf{x}) \triangleq \frac{f(\mathbf{x} + \epsilon \mathbf{e}_i) - f(\mathbf{x} - \epsilon \mathbf{e}_i)}{2\epsilon}, \tag{3}$$

where $\epsilon > 0$ is the finite difference step size.

The update rule for the gradient estimate is then given by

$$\hat{\mathbf{g}}^{(k+1)} = \hat{\mathbf{g}}^{(k)} + \left( \tilde{\nabla}_{i^{(k)}} f(\mathbf{x}_t) - \mathbf{e}_{i^{(k)}}^{\top} \hat{\mathbf{g}}^{(k)} \right) \mathbf{e}_{i^{(k)}}. \tag{4}$$

Intuitively, Eq. (4) replaces the stale gradient estimate stored at coordinate $i^{(k)}$ with the newly computed finite difference value, ensuring that $\hat{\mathbf{g}}$ always contains the most recent information available for each coordinate.

## 4. Methodology: Coherent Coordinate Descent

In this section, we present Coherent Coordinate Descent (CoCD), a deterministic zeroth-order optimization framework designed to bridge the gap between heuristic efficiency and theoretical rigor.

### 4.1. The CoCD Algorithm and Unifying Momentum

At the core of CoCD is a dense gradient buffer $\hat{\mathbf{g}} \in \mathbb{R}^n$, which maintains an estimate of the global gradient. Unlike randomized methods that construct a new estimate from scratch at every step, CoCD performs incremental updates.

**The Update Cycle.** Let $\hat{\mathbf{g}}_t$ denote the gradient buffer used at optimization step $t$. In each step, given a compute budget of $B$ function queries, CoCD updates a subset of coordinates. The process consists of two phases: a *temporal decay* phase and a *refresh* phase.

First, we apply a global momentum decay factor $\gamma \in [0, 1]$ to the entire buffer:

$$\hat{\mathbf{g}}_{t-1}^{\text{decay}} = \gamma \cdot \hat{\mathbf{g}}_{t-1}. \tag{5}$$

Next, we sequentially update $B$ coordinates. For each sub-step $k = 1, \ldots, B$, we select the coordinate index $i$ according to the cyclic schedule defined in Sec. 3.2, compute the fresh finite difference $\tilde{\nabla}_i f(\mathbf{x}_t)$, and overwrite the corresponding entry in the buffer:

$$\hat{\mathbf{g}}_t[i] = \tilde{\nabla}_i f(\mathbf{x}_t). \tag{6}$$

For all other coordinates $j \neq i$ that were not selected in this batch, the buffer retains its decayed value:

$$\hat{\mathbf{g}}_t[j] = \hat{\mathbf{g}}_{t-1}^{\text{decay}}[j].$$

Finally, the model parameters are updated using this hybrid gradient estimate:

$$\mathbf{x}_{t+1} = \mathbf{x}_t - \alpha \cdot \hat{\mathbf{g}}_t, \tag{7}$$

where $\alpha$ is the learning rate.

**Unification of CCD and CoCD.** The parameter $\gamma$ serves a dual purpose: it acts as a stabilizer for the gradient history in practice and, theoretically, provides a unifying view of coordinate-wise optimization methods.

- **Case $\gamma = 1.0$ (Standard CoCD):** The buffer acts as a perfect memory. Stale gradient estimates are preserved indefinitely until they are cyclically refreshed. This maximizes information utilization, effectively performing full-gradient descent with a time-lagged gradient estimate.

- **Case $\gamma = 0.0$ (Classical BCCD):** The buffer is effectively cleared at the start of each step. The gradient estimate $\hat{\mathbf{g}}$ contains non-zero entries *only* for the $B$ coordinates updated in the current step. This recovers the standard BCCD behavior, where no stale information is used.

- **Case $0 < \gamma < 1$ (Fading Memory):** This setting provides a smooth interpolation. It allows the algorithm to leverage historical information while exponentially suppressing very old (and potentially invalid) gradient estimates, offering robustness in highly non-convex or rapidly changing landscapes.

### 4.2. Efficient Implementation: The Flattened FIFO Buffer

Implementing CoCD for training neural networks presents a unique engineering challenge: the parameters $\mathbf{x}$ are typically structured as a collection of tensors with varying shapes (e.g., 4D convolution kernels, 2D weight matrices), whereas the coordinate descent logic operates naturally on a flat vector space. Naively reshaping tensors at every step would incur prohibitive memory copy overhead.

To address this, we implement a **virtualized flattening mechanism** with $O(1)$ overhead. For the detailed algorithmic implementation, we refer readers to the pseudocode in Appendix D.

1. **Global Gradient Buffer:** We allocate a single contiguous 1D tensor (the FIFO buffer) of size $m$, where $m$ is the user-defined memory budget. Typically $m = n$ (total parameter count), but it can be smaller for memory-constrained devices.

2. **State Pointers:** We maintain a set of integer pointers—'cur_param_idx', 'cur_weight_idx', and 'cur_grad_idx'—that track the mapping between the flat buffer indices and the structured model parameters.

3. **In-Place Operations:** During the *optimization* phase (descent), instead of reconstructing the gradient tensors, we create temporary views of the flat buffer that match the shapes of the target parameters. This allows us to apply the update $\mathbf{x} \leftarrow \mathbf{x} - \alpha\hat{\mathbf{g}}$ completely in-place, without allocating additional memory for gradients.

This design ensures that CoCD maintains a strict memory footprint of exactly one copy of the model parameters (for the gradient buffer), which is significantly more memory-efficient than adaptive first-order methods like Adam (which require two copies for moments) or random subspace methods that often require storing large projection matrices.

## 5. Theoretical Analysis

In this section, we provide a rigorous analysis of CoCD. We analyze the behaviors standard of CoCD (with momentum $\gamma = 1$ and memory budget $m = n$) in § 5.1 and § 5.2. We first formalize the notion of *local coherence* and *implicit smoothing*, then derive upper bounds on the gradient approximation error induced by staleness in § 5.1. Building on this result, we analyze CoCD's global convergence under certain assumptions in § 5.2. Finally, in § 5.3, we take varying $\gamma$ and $m$ into account and examine their impact on CoCD's stability.

### 5.1. Approximation Error with Implicit Smoothing

We begin by analyzing the quality of the gradient estimate $\hat{\mathbf{g}}_t$ produced by CoCD relative to the true gradient. To isolate the error introduced specifically by *staleness*, we use the fact

$$\|\hat{\mathbf{g}}_t - \nabla f(\mathbf{x}_t)\| \leq \underbrace{\left\|\hat{\mathbf{g}}_t - \tilde{\nabla}^\epsilon f(\mathbf{x}_t)\right\|}_{\text{staleness error}} + \underbrace{\left\|\tilde{\nabla}^\epsilon f(\mathbf{x}_t) - \nabla f(\mathbf{x}_t)\right\|}_{\text{finite difference error}} \quad (8)$$

where $\tilde{\nabla}^\epsilon f$ denote the gradient calculated using finite difference with an interval $\epsilon$. We thus only analyze the first term for the two reasons: 1). Bounding the error introduced by finite difference is beyond the scope of this work, and we refer readers to Berahas et al. (2022) for detailed analyses of the error. 2). Since the staleness error is accumulative over the optimization trajectory while the finite difference error is "point-wise", the first term dominates the total error.

**Defining Coherence.** The central premise of our method is that gradients do not change arbitrarily between successive optimization steps. We formalize this assumption through local smoothness of the optimization trajectory.

**Definition 5.1** (Local Coherence)**.** Let $\mathcal{T}_{t,k} = \{\mathbf{x}_{t-k}, \ldots, \mathbf{x}_t\}$ denote the sequence of the most recent $k + 1$ iterates. We say the optimization trajectory is *locally coherent* over $\mathcal{T}_{t,k}$ if the objective function $f$ is $L$-smooth within the convex hull of $\mathcal{T}_{t,k}$. We also define the associated local step bound

$$\delta_{t,k} \geq \max_{j \in \{t-k+1, \cdots, t\}} \|\mathbf{x}_j - \mathbf{x}_{j-1}\|.$$

When the window is clear from context, we write $\delta$.

**Defining Implicit Smoothing.** Finite-difference gradients implicitly smooth the optimization landscape by replacing each partial derivative with a local coordinate-wise average. For example, central finite-difference approximation satisfies

$$\tilde{\nabla}_i^\epsilon f(\mathbf{x}) = \frac{1}{2\epsilon} \int_{-\epsilon}^{\epsilon} \nabla_i f(\mathbf{x} + u\mathbf{e}_i) \, du,$$

by the Fundamental Theorem of Calculus, where $\mathbf{e}_i$ is the $i$-th standard basis vector. Thus, $\epsilon$ determines the averaging scale: larger values of $\epsilon$ average gradient information over a wider coordinate-wise neighborhood and can attenuate small-scale oscillations in highly nonconvex landscapes. We define the uniform coordinate-wise smoothness constant $L_\epsilon$ as follows to formalize the notion of implicit smoothing.

**Definition 5.2** (Uniform Coordinate-wise Smoothness with Implicit Smoothing)**.** Let $f$ be an $L$-smooth function. We define the uniform coordinate-wise Lipschitz constant $L_\epsilon$ of the finite-difference gradient $\tilde{\nabla}^\epsilon f(\mathbf{x})$ as

$$L_\epsilon := \max_{i \in [n]} \sup_{\mathbf{x} \neq \mathbf{y}} \frac{|\tilde{\nabla}_i^\epsilon f(\mathbf{x}) - \tilde{\nabla}_i^\epsilon f(\mathbf{y})|}{\|\mathbf{x} - \mathbf{y}\|}.$$

This definition corresponds to the $2 \to \infty$ induced norm of the Hessian, whereas standard smoothness corresponds to the $2 \to 2$ induced norm, i.e., the spectral norm. It serves to construct the bounds below that treat blocks of coordinates separately.

**Theorem 5.3** (Approximation Error Bound)**.** *Suppose* $f : \mathbb{R}^n \to \mathbb{R}$ *is locally coherent with the smoothness parameter* $L$. *At step $t$, let* $\hat{\mathbf{g}}_t$ *denote the gradient estimate maintained by CoCD with a compute budget of $B$ queries per step and a finite difference interval $\epsilon$. Let* $K = \lfloor \frac{n}{B} \rfloor$ *and* $r = n \mod B$. *Then the approximation error satisfies:*

$$\left\|\hat{\mathbf{g}}_t - \tilde{\nabla}^\epsilon f(\mathbf{x}_t)\right\| \leq \frac{L_\epsilon \delta}{2} \left(BK(K-1) + 2rK\right)$$
$$\leq \frac{L\delta}{2} \left(BK(K-1) + 2rK\right). \quad (9)$$

*Proof.* See Appendix A. □

**Interpretation.** The error bound is governed by the product $L_\epsilon \cdot \delta$. This formalizes a key insight: approximation quality degrades only when the objective is highly non-smooth (large $L_\epsilon$) or when the optimizer takes excessively large steps (large $\delta$). An appropriately large $\epsilon$ can reduce $L_\epsilon$, and thus improve the smoothness of the optimization landscape. Consequently, accurate gradient estimates can be maintained even in the presence of stale information. An empirical verification of the bound is given in Appendix B.

We highlight three corollaries that characterize the behavior of CoCD under different compute budgets:

**Corollary 5.4** (Worst-Case Performance ($B = 1$)). *With a minimal budget of $B = 1$, the approximation error scales as $O(n^2 L_\epsilon \delta)$. To maintain the $O(1/n)$ accuracy typically required for stable descent, the step size must scale such that $\delta = O(n^{-3})$.*

**Corollary 5.5** (Efficient Regime ($B = \sqrt{n}$)). *With a budget of $B = \sqrt{n}$, the error bound improves to $O(n^{1.5} L_\epsilon \delta)$. This permits more aggressive step sizes ($\delta = O(n^{-2})$) while maintaining approximation fidelity on the order of $O(1/\sqrt{n})$.*

**Corollary 5.6** (Fallback to Finite-Differenced Gradient ($B = n$)). *When the budget covers all coordinates ($B = n$), we have $\lfloor n/B \rfloor = 1$, and the right-hand side of Theorem 5.3 becomes zero. In this case, CoCD recovers the exact finite difference gradient, confirming that it generalizes standard full-gradient methods.*

### 5.2. Global Convergence Analysis

We now establish the *theoretical ceiling* of CoCD by demonstrating that under the exact same structural assumptions where gradient descent achieves linear convergence, stale gradients also properly achieve linear convergence.

**Assumption 5.7** ($L$-Smoothness). The gradient of $f$ is Lipschitz continuous, i.e.,

$$\|\nabla f(\mathbf{x}) - \nabla f(\mathbf{y})\| \leq L\|\mathbf{x} - \mathbf{y}\| \quad \text{for all } \mathbf{x}, \mathbf{y}.$$

**Assumption 5.8** (Polyak-Łojasiewicz (PŁ) Condition). There exists $\mu > 0$ such that

$$\|\nabla f(\mathbf{x})\|^2 \geq 2\mu\big(f(\mathbf{x}) - f(\mathbf{x}^*)\big),$$

where $\mathbf{x}^*$ is a global minimizer.

**Assumption 5.9** (Sufficiently Small Finite Difference Interval). We assume the finite difference interval $\epsilon \to 0$ for approximating the gradients of function $f$.

Under these assumptions, we establish a linear convergence rate for CoCD, comparable to that of first-order methods.

**Theorem 5.10** (Linear Convergence of CoCD). *Let Assumptions 5.7, 5.8, and 5.9 hold, and let $\alpha$ denote the learning rate. Define the staleness factor $\tau = n/B - 1$. Then, after $t$ iterations,*

$$f(\mathbf{x}_t) - f(\mathbf{x}^*) \leq \left(1 - \frac{2\mu C_1}{C_2}\right)^t \big(f(\mathbf{x}_0) - f(\mathbf{x}^*)\big), \quad (10)$$

*provided the constants*

$$C_1 = \frac{1}{\alpha} - \frac{L}{2}(1 + n\tau), \qquad C_2 = \frac{1}{\alpha^2} + \frac{Ln\tau}{\alpha} + \frac{L^2 n^2 \tau^2}{4}$$

*are positive.*

*Proof.* See Appendix C. □

In practice, convergence can be much faster than what the theorem suggests. We refer readers to Remark C.2 in Appendix C for detailed explanations.

**Corollary 5.11** (Equivalence to Gradient Descent). *In the full-budget case ($B = n$), we have $\tau = 0$, and the rate reduces to $1 - 2\mu\alpha(1 - \frac{L\alpha}{2})$, recovering the PŁ-type gradient descent rate.*

Next, we establish a weaker convergence guarantee even without the PŁ condition.

**Theorem 5.12** (Gradient Estimate Convergence). *Under Assumption 5.7, for a bounded function $f$, the sequence of gradient estimates satisfies*

$$\|\hat{\mathbf{g}}_t - \hat{\mathbf{g}}_{t-1}\| \to 0 \quad \text{as } t \to \infty.$$

*Proof.* See Appendix C. □

This result shows that the maintained gradient estimate stabilizes asymptotically.

**Impact of Implicit Smoothing on Convergence.** Adapting our proofs to the framework proposed in Karimi et al. (2016) for analyzing proximal gradient descent with the error bounds of finite-differenced gradients obtained in Berahas et al. (2022), we can show linear convergence and gradient estimate convergence still hold when $\epsilon > 0$. For linear convergence, we will have:

$$f(\mathbf{x}_t) - f(\mathbf{x}^*) \leq \left(1 - \frac{2\mu C_1}{C_2}\right)^t \big(f(\mathbf{x}_0) - f(\mathbf{x}^*)\big) + err(\epsilon),$$
(11)

where

$$C_1 \approx \frac{1}{\alpha} - \frac{L_\epsilon}{2}(1 + n\tau), \qquad C_2 \approx \frac{1}{\alpha^2} + \frac{L_\epsilon n\tau}{\alpha} + \frac{L_\epsilon^2 n^2 \tau^2}{4},$$

and $err(\epsilon)$ is a polynomial of $\epsilon$ accounting for the bias caused by finite difference.

An appropriately larger $\epsilon$ can reduce $L_\epsilon$, achieving better convergence rate, though it could potentially increase $err(\epsilon)$, pushing the result away from the true global optima. Implicit smoothing with smaller $L_\epsilon$ also enables the stability condition $C_1 > 0$ to be satisfied for larger learning rate $\alpha$. Our evaluations in § 6.1 corroborates with the analyses.

### 5.3. Stability Analysis

We now analyze how varying compute budgets, momentum, and memory budgets influence stability.

According to Theorem 5.10, we need $C_1 > 0$ to ensure stable descent. Thus the learning rate must satisfy $\alpha < \frac{2}{L(1+n\tau)}$. This highlights a trade-off: larger problem dimensions ($n$) or increased staleness ($\tau$) require smaller step sizes for stability. Increasing the compute budget $B$ reduces $\tau$, enabling larger learning rates and faster convergence.

Applying fading memory $0 < \gamma < 1$ for approximating gradients is equivalent to applying geometrically decaying learning rates $\{\alpha, \alpha\gamma, ..., \alpha\gamma^\tau\}$ to $\tau + 1$ blocks of coordinates from fresh to stale during descent. Momentum thus exponentially suppresses the weight of older gradients, resulting in stabler descent. Memory budget $m$ acts equivalently as a hard truncation to the decaying learning rates, setting zero learning rates for all blocks with staleness larger than $\frac{m}{B}$. We denote the effective learning rate to be $\alpha_{\text{eff}}(\gamma, m) \ll \alpha$. We then need $\alpha_{\text{eff}}(\gamma, m) < \frac{2}{L(1+n\tau)}$ for stable training. The actual learning rate can be much larger than the bound.

## 6. Evaluation

In this section, we evaluate the performance of CoCD on three benchmark datasets: SARCOS (Vijayakumar & Schaal, 2000) (regression), MNIST (LeCun et al., 1998), and CIFAR-10 (Krizhevsky et al., 2009) (classification). We compare CoCD against standard Block Cyclic Coordinate Descent (BCCD, i.e., $\gamma = 0$) and Stochastic Gradient Descent (SGD). We further conduct ablation studies to analyze the impact of the smoothing radius $\epsilon$, momentum coefficient $\gamma$, compute budget $B$, and memory budget $M$. Finally, we compare CoCD against randomized zeroth-order baselines under matched compute budgets.

In addition, we demonstrate in Appendix E that CoCD can scale up stably to Resnet-20 (with $\approx 270$k paramters) training with sufficiently large compute budget in the given time limit.

All experiments were implemented in PyTorch (Paszke et al., 2019) and conducted on a workstation equipped with a single NVIDIA RTX PRO 6000 Blackwell GPU.

### 6.1. Main Results: Convergence and Accuracy

We first report the training performance of CoCD relative to the baselines. (1) **Models:** For SARCOS, we use a 5-layer MLP with approximately 13k parameters. For MNIST and CIFAR-10, we employ a CNN with three convolutional layers followed by one fully connected layer, totaling approximately 20k parameters. (2) **Baselines:** We compare against BCCD ($\gamma = 0$) to isolate the effect of gradient reuse via momentum. For BCCD, we use a small smoothing radius $\epsilon = 10^{-6}$, which empirically yields the strongest performance for standard coordinate descent. See Figure 2 on how larger smoothing radii improve the performance of CoCD while degrade that of BCCD. SGD is included solely as a first-order oracle reference, using exact gradients and momentum fixed at 0.9. (3) **Training settings:** All methods are trained for 50 epochs. We use a batch size of 64 for regression and 128 for classification. To ensure a fair comparison, the learning rate $\alpha$ and weight decay are aligned across CoCD, BCCD, and SGD. Specifically, we use a weight decay of $10^{-4}$ for SARCOS and MNIST, and $5 \times 10^{-4}$ for CIFAR-10. The CoCD-specific hyperparameters are summarized in Table 1. The quantitative results are summarized in Table 2.

*Table 1.* Hyperparameter settings for CoCD. Notation: learning rate ($\alpha$), momentum ($\gamma$), smoothing radius ($\epsilon$), and compute budget ($B$). The budget is reported as the raw number of queries followed by the percentage of total trainable parameters in parentheses. For BCCD, we use $\epsilon = 10^{-6}$.

| Dataset | LR ($\alpha$) | Momentum ($\gamma$) | Smoothing ($\epsilon$) | Budget ($B$) |
|---|---|---|---|---|
| SARCOS | 0.001 | 1.0 | 1.0 | 64 ($\approx$0.5%) |
| MNIST | 0.01 | 0.99 | 0.1 | 256 ($\approx$1.3%) |
| CIFAR-10 | 0.01 | 0.99 | 0.1 | 1024 ($\approx$4.2%) |

*Table 2.* Final loss (SARCOS) and accuracy (MNIST, CIFAR-10). SGD uses exact gradients and serves as an oracle reference. Training time is reported in seconds per epoch.

| Dataset | Method | Final Metric | Time (s/epoch) |
|---|---|---|---|
| **SARCOS** | SGD (Oracle) | *Loss:* 5.38 | – |
| | BCCD ($\gamma = 0$) | 188.73 | 6.23 |
| | **CoCD (Ours)** | **31.18** | 6.12 |
| **MNIST** | SGD (Oracle) | *Acc:* 99.21% | – |
| | BCCD ($\gamma = 0$) | 27.03% | $\approx$ 44.42 |
| | **CoCD (Ours)** | **95.48%** | $\approx$ 44.56 |
| **CIFAR-10** | SGD (Oracle) | *Acc:* 62.51% | – |
| | BCCD ($\gamma = 0$) | 10.13% | $\approx$ 77.01 |
| | **CoCD (Ours)** | **45.08%** | $\approx$ 77.03 |

**Analysis.** As shown in Table 2, CoCD consistently outperforms BCCD across all tasks while substantially closing the gap to the first-order oracle. These gains are achieved with negligible additional computational overhead, confirming that the momentum-based gradient reuse is effectively free.

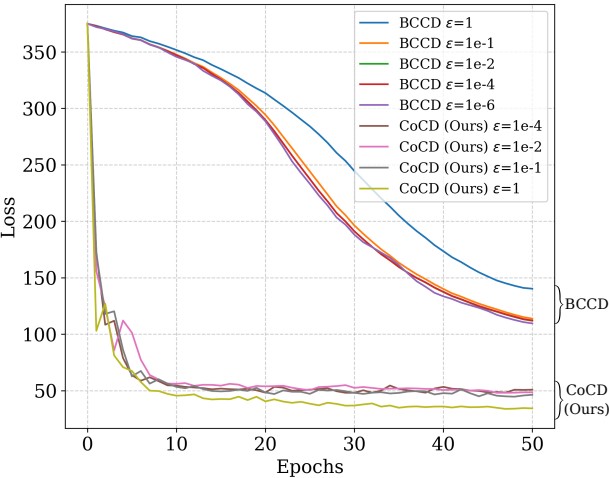

*Figure 2.* **CoCD accelerates derivative-free training on the SAR-COS robotics dataset.** We train a 5-layer MLP ($\approx$ 12k parameters) using our proposed Coherent Coordinate Descent (CoCD) versus standard Block Cyclic Coordinate Descent (BCCD) with a fixed query budget ($B \approx 2\%$ of all parameters). The results highlight two key advantages: (1) **Performance Gap:** CoCD (lower cluster) dramatically outperforms BCCD (upper cluster) in validation loss. (2) **The Smoothing Anomaly:** While large smoothing radii ($\epsilon = 1$) degrade BCCD performance (blue line, top), they act as a stabilizer for CoCD (yellow line, bottom), allowing it to achieve the lowest final loss. This indicates CoCD's unique ability to leverage landscape smoothing for robust optimization.

The effect is particularly pronounced on CIFAR-10: while BCCD fails to escape the random-guessing regime, CoCD recovers meaningful representations and reaches 45.08% accuracy, indicating that coherence-based updates provide crucial robustness in high-dimensional, non-convex settings.

### 6.2. Ablation Studies

We next study the influence of key hyperparameters. Experiments focus on two representative settings: an MLP on SARCOS and a CNN on MNIST. Unless otherwise stated, we use the following defaults: (1) **SARCOS:** $\epsilon = 1.0$, $\gamma = 1.0$, $B = 64$, $M = \frac{m}{n} = 1.0$; (2) **MNIST:** $\epsilon = 1.0$, $\gamma = 1.0$, $B = 256$, $M = \frac{m}{n} = 1.0$. The results can be seen in Figure 3.

**Impact of Smoothing Radius ($\epsilon$).** On SARCOS, performance improves monotonically with increasing $\epsilon$, as larger smoothing filters out high-frequency noise in a highly non-convex landscape. On MNIST, the trend is more nuanced: smaller $\epsilon$ yields higher final accuracy when the compute budget is large, but larger $\epsilon$ produces noticeably smoother training curves and improved stability.

**Impact of Momentum ($\gamma$).** Momentum is the dominant factor governing convergence speed and robustness. Setting $\gamma = 0$ yields the slowest convergence, while increasing $\gamma$ consistently accelerates training. In an extreme sparsity stress test on SARCOS ($B = 1$), tuning $\gamma = 0.95$ and $\epsilon = 1.0$ allows CoCD to converge to a loss of 68.64, whereas BCCD ($\gamma = 0$) stagnates near 188.0 even with $B = 64$, demonstrating that gradient history is essential for coherent descent.

**Impact of Compute ($B$) and Memory ($M$).** We observe a clear stability threshold with respect to $B$: for the MLP, $B = 64$ is the minimum budget required to rapidly escape the initial plateau, while larger budgets yield diminishing returns. Reducing the memory budget $M$ introduces additional noise but does not prevent convergence. Even with $M = 0.25$, CoCD maintains competitive performance, validating the memory-efficient design described in Section 4.2.

### 6.3. Comparison with Random-Update Baselines

Finally, we compare CoCD against randomized zeroth-order baselines (SPSA and ZO-SGD (Ghadimi & Lan, 2013)) under a matched compute budget per iteration. The results can be seen in Figure 4.

**Discussion.** On classification tasks, baselines match CoCD's performance in terms of the final accuracy achieved. However, for the regression task on SARCOS, as shown in Figure 4, baselines using randomized updates diverges half way while CoCD, which tends to exhibit diffusive behavior due to isotropic sampling, stabilizes. By exploiting coordinate-wise structure and temporal coherence, CoCD constructs both informative and stable update directions.

Especially for ZO-SGD which relies on explicit smoothing, it requires the computational overhead of drawing dense Gaussian samples at every single step. CoCD leverages implicit smoothing deterministically, completely bypassing this random sampling overhead for improved wall-clock speed. CoCD is almost 2x faster than ZO-SGD in our experiment. (`8.1s` vs. `15.7s` per episode on average).

## 7. Conclusion and Future Work

In this work, we introduced Coherent Coordinate Descent (CoCD), a deterministic zeroth-order optimization framework that offers an alternative to purely randomized gradient estimation. CoCD exploits the temporal continuity of optimization trajectories by reusing stale gradient information in a structured and lightweight manner, enabling efficient descent without stochastic sampling.

**Theoretical Contributions.** We established a rigorous theoretical foundation for CoCD by proving its equivalence to Block Cyclic Coordinate Descent (BCCD) with stale warm

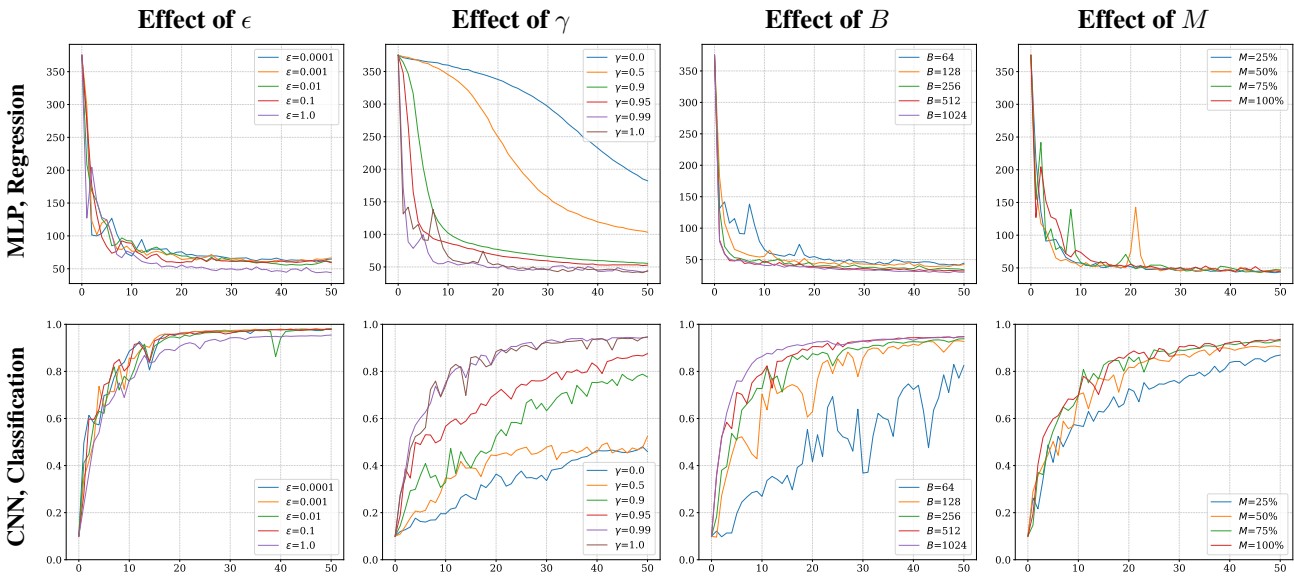

*Figure 3.* Ablation results on SARCOS (top, y-axis is validation loss) and MNIST (bottom, y-axis is validation accuracy).

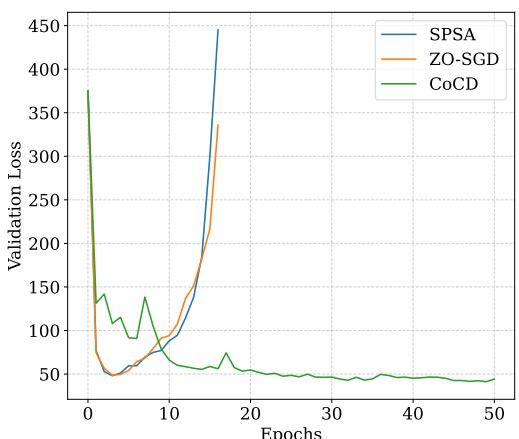

*Figure 4.* Validation loss comparison among CoCD, ZO-SGD and SPSA under equal function-evaluation budgets (64). In terms of wall clock time, CoCD is almost 2x faster than ZO-SGD. (`8.1s` vs. `15.7s` per episode on average).

starts. Under the Polyak-Łojasiewicz (PŁ) condition, we showed that CoCD achieves linear convergence despite operating on stale gradient estimates. Our analysis further derived explicit approximation error bounds that account for both gradient staleness and the finite difference interval $\epsilon$. These bounds provide a theoretical explanation for the empirically observed benefit of larger $\epsilon$ values: coarser finite differences implicitly smooth the optimization landscape by reducing the effective Lipschitz constant, thereby improving stability in non-convex settings.

**Empirical Findings.** We evaluated CoCD on regression and classification benchmarks, including evaluations on MLPs, LeNet-style CNNs, and Resnet-20 (up to ≈270k parameters). The results consistently support the theory: (1) CoCD substantially outperforms standard BCCD, confirming that reusing stale gradient history is significantly more effective than discarding it; (2) larger smoothing radii $\epsilon$ improve training stability and often lead to better final performance, validating the predicted implicit smoothing effect; and (3) the momentum parameter $\gamma$ plays a critical stabilizing role, allowing CoCD to interpolate smoothly between aggressive history reuse and the conservative behavior of BCCD.

**Limitations and Future Directions.**

At present, CoCD is most effective in small-to-medium scale models. As the parameter count grows beyond a certain threshold, achieving sufficiently accurate coordinate-wise gradient estimates requires a compute budget that becomes impractically large. Future work will focus on addressing these scaling challenges through block-wise parallelization, adaptive compute and memory budgeting, and structured coordinate selection strategies that prioritize informative parameter groups. For system-level implementations, substituting DeepZero's core sparse-CGE estimator with CoCD is a highly promising trajectory for scaling up. Similarly, adapting CoCD for LLM fine-tuning tasks to compare its efficiency against MeZO (Malladi et al., 2023) (which is based on ZO-SGD) presents an exciting direction as well. Additional directions include extending the theoretical analysis beyond the PŁ condition, developing adaptive schemes for selecting $\epsilon$ and $\gamma$ based on local landscape properties, and examining the gradient-reuse strategy in optimizers beyond zeroth-order, such as AdamW (Loshchilov & Hutter, 2019) and Muon (Jordan et al., 2024).

## Impact Statement

This work introduces Coherent Coordinate Descent (CoCD), a deterministic zeroth-order optimization method for settings where gradient information is unavailable or expensive to obtain. By enabling efficient optimization using only function evaluations and limited memory, CoCD may benefit applications in edge AI, embedded systems, and other resource-constrained environments.

The proposed method does not introduce new classes of societal risk beyond those inherent to machine learning optimization techniques. CoCD does not increase model expressivity, autonomy, or data access, and does not enable new forms of data collection or surveillance. As with other optimization methods, misuse remains possible if applied irresponsibly.

Finally, CoCD is currently most effective in small-to-medium scale models and memory-constrained regimes. Future extensions to larger systems should be evaluated carefully to ensure efficiency gains do not come at the expense of stability or robustness. Overall, we view CoCD as a focused methodological contribution with predominantly positive potential impact when used responsibly.

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

## A. Proofs of Error Bounds

**Theorem A.1** (Approximation Error Bound). *Suppose $f : \mathbb{R}^n \to \mathbb{R}$ is locally coherent with the smoothness parameter $L$. At step $t$, let $\hat{\mathbf{g}}_t$ denote the gradient estimate maintained by CoCD with a compute budget of $B$ queries per step and a finite difference interval $\epsilon$. Let $K = \lfloor \frac{n}{B} \rfloor$ and $r = n \bmod B$. Then the approximation error satisfies:*

$$
\begin{aligned}
\left\| \hat{\mathbf{g}}_t - \tilde{\nabla}^\epsilon f(\mathbf{x}_t) \right\| &\leq \frac{L_\epsilon \delta}{2} \left( BK(K-1) + 2rK \right) \\
&\leq \frac{L\delta}{2} \left( BK(K-1) + 2rK \right).
\end{aligned}
\tag{12}
$$

*Proof.* We begin by proving that $L_\epsilon \leq L$. For the central finite-difference gradient,

$$
\nabla_i^\epsilon f(\mathbf{x}) = \frac{f(\mathbf{x} + \epsilon \mathbf{e}_i) - f(\mathbf{x} - \epsilon \mathbf{e}_i)}{2\epsilon},
$$

where $\mathbf{e}_i$ is the $i$-th standard basis vector. By the Fundamental Theorem of Calculus,

$$
\nabla_i^\epsilon f(\mathbf{x}) = \frac{1}{2\epsilon} \int_{-\epsilon}^{\epsilon} \nabla_i f(\mathbf{x} + u\mathbf{e}_i) \, du.
$$

Hence, for any $\mathbf{x}, \mathbf{y} \in \mathbb{R}^n$,

$$
\begin{aligned}
|\nabla_i^\epsilon f(\mathbf{x}) - \nabla_i^\epsilon f(\mathbf{y})| &\leq \frac{1}{2\epsilon} \int_{-\epsilon}^{\epsilon} |\nabla_i f(\mathbf{x} + u\mathbf{e}_i) - \nabla_i f(\mathbf{y} + u\mathbf{e}_i)| \, du \\
&\leq \frac{1}{2\epsilon} \int_{-\epsilon}^{\epsilon} \|\nabla f(\mathbf{x} + u\mathbf{e}_i) - \nabla f(\mathbf{y} + u\mathbf{e}_i)\| \, du \\
&\leq \frac{1}{2\epsilon} \int_{-\epsilon}^{\epsilon} L\|\mathbf{x} - \mathbf{y}\| \, du \\
&= L\|\mathbf{x} - \mathbf{y}\|.
\end{aligned}
\tag{13}
$$

Therefore,

$$
\sup_{\mathbf{x} \neq \mathbf{y}} \frac{|\nabla_i^\epsilon f(\mathbf{x}) - \nabla_i^\epsilon f(\mathbf{y})|}{\|\mathbf{x} - \mathbf{y}\|} \leq L
$$

for every $i \in [n]$. Taking the maximum over $i$ gives

$$
L_\epsilon \leq L.
$$

The same argument applies to forward and backward finite differences by replacing the averaging interval $[-\epsilon, \epsilon]$ with $[0, \epsilon]$ or $[-\epsilon, 0]$, respectively.

The gradient estimate $\hat{\mathbf{g}}_t$ maintained by CoCD is a composite of partial derivatives computed at varying degrees of staleness. We partition the $n$ coordinates based on the time lag $\tau$ since their last update.

Since $B$ coordinates are updated per step, we have $K = \lfloor n/B \rfloor$ full blocks and one remainder block of size $r$. At step $t$:

- For each lag $\tau \in \{0, 1, \ldots, K-1\}$, there are exactly $B$ coordinates that were last updated at step $t - \tau$.

- The remaining $r$ coordinates were last updated at step $t - K$.

Let $g_i(\mathbf{x}) = \tilde{\nabla}_i^\epsilon f(\mathbf{x})$. By the uniform coordinate-wise Lipschitz continuity of $\tilde{\nabla}^\epsilon f$, the error for a single coordinate $i$ updated $\tau$ steps ago is bounded by:

$$
|\hat{g}_{t,i} - g_i(\mathbf{x}_t)| = |g_i(\mathbf{x}_{t-\tau}) - g_i(\mathbf{x}_t)| \leq L_\epsilon \|\mathbf{x}_{t-\tau} - \mathbf{x}_t\| \leq L_\epsilon \tau \delta.
\tag{14}
$$

Summing this error over all $n$ coordinates:

$$
\begin{aligned}
\left\| \hat{\mathbf{g}}_t - \tilde{\nabla}^\epsilon f(\mathbf{x}_t) \right\| &\leq \sum_{\tau=0}^{K-1} \underbrace{B \cdot (L_\epsilon \tau \delta)}_{\text{Block } \tau} + \underbrace{r \cdot (L_\epsilon K \delta)}_{\text{Remainder}} \\
&= L_\epsilon \delta B \sum_{\tau=0}^{K-1} \tau + L_\epsilon \delta r K \\
&= L_\epsilon \delta B \frac{K(K-1)}{2} + L_\epsilon \delta r K \\
&= \frac{L_\epsilon \delta}{2} \left( BK(K-1) + 2rK \right) \\
\\
&\leq \frac{L\delta}{2} \left( BK(K-1) + 2rK \right).
\end{aligned}
\tag{15}
$$

$\square$

## B. Empirical Validation of Error Bounds

To empirically verify Theorem 5.3, we conducted an experiment plotting the average $\|\hat{\mathbf{g}}_t - \tilde{\nabla}^\epsilon f(\mathbf{x}_t)\|$ with respect to the compute budget $B$ during the training of a feed-forward network with $\approx 4k$ parameters on the SARCOS dataset.

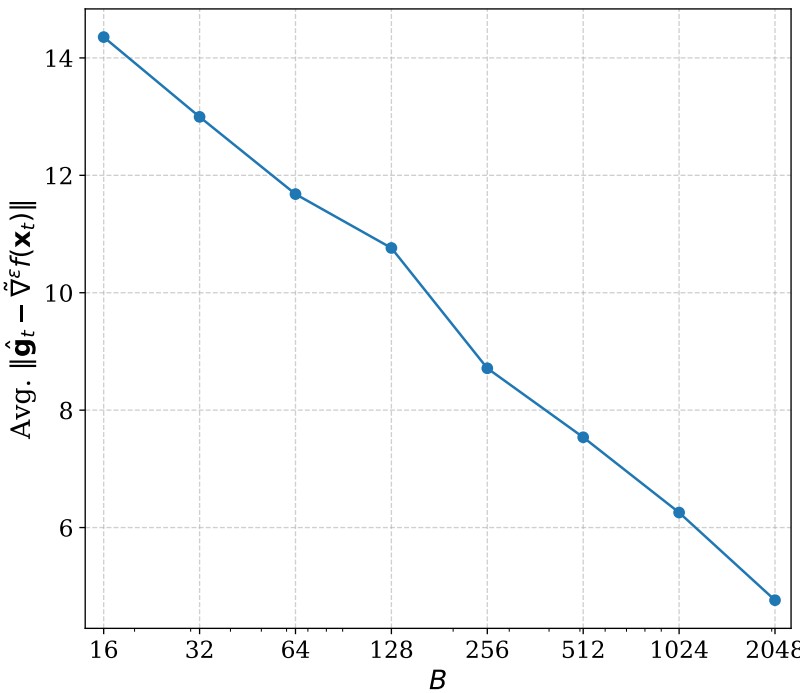

*Figure 5.* Average gradient approximation error $\|\hat{\mathbf{g}}_t - \tilde{\nabla}^\epsilon f(\mathbf{x}_t)\|$ with respect to the different compute budgets $B$ ranging from 16 to 2048.

As shown in Fig. 5, the x-axis is log-scale and the error curve linearly fits a straight line $y = ax + b$ (where $a < 0$). This exactly corroborates the theoretical bound derived in our theorem: as the compute budget increases, the deterministic approximation error decreases predictably.

# C. Proofs of Convergence of CoCD

**Theorem C.1** (Linear Convergence of CoCD). *Let Assumptions 5.7, 5.8, and 5.9 hold, and let $\alpha$ denote the learning rate. Define the staleness factor $\tau = n/B - 1$. Then, after $t$ iterations,*

$$f(\mathbf{x}_t) - f(\mathbf{x}^*) \leq \left(1 - \frac{2\mu C_1}{C_2}\right)^t (f(\mathbf{x}_0) - f(\mathbf{x}^*)), \tag{16}$$

*provided the constants*

$$C_1 = \frac{1}{\alpha} - \frac{L}{2}(1 + n\tau), \qquad C_2 = \frac{1}{\alpha^2} + \frac{Ln\tau}{\alpha} + \frac{L^2 n^2 \tau^2}{4}$$

*are positive.*

*Proof.* By the $L$-smoothness of $f$ (Assumption 5.7), we have the standard inequality:

$$f(\mathbf{x}_t) - f(\mathbf{x}_{t-1}) \leq \nabla f(\mathbf{x}_{t-1})^\top (\mathbf{x}_t - \mathbf{x}_{t-1}) + \frac{L}{2}\|\mathbf{x}_t - \mathbf{x}_{t-1}\|^2. \tag{17}$$

The CoCD update rule is given by $\mathbf{x}_t = \mathbf{x}_{t-1} - \alpha\hat{\mathbf{g}}_{t-1}$, which implies:

$$\hat{\mathbf{g}}_{t-1} = \frac{1}{\alpha}(\mathbf{x}_{t-1} - \mathbf{x}_t). \tag{18}$$

We define the gradient approximation error (correction term) at step $t-1$ as:

$$\mathbf{c}_{t-1} = \nabla f(\mathbf{x}_{t-1}) - \hat{\mathbf{g}}_{t-1}. \tag{19}$$

Under Assumption 5.9, the error induced by finite differences is zero. We then invoke Theorem 5.3 to directly bound the correction term. Assuming for simplicity that $n$ is divisible by the budget $B$ (i.e., $r = 0$), we have $K = n/B$. Substituting $\tau = n/B - 1 = K - 1$ into the bound from Theorem 5.3 yields:

$$\|\mathbf{c}_{t-1}\| \leq \frac{L\delta}{2}\left(B \cdot \frac{n}{B}\left(\frac{n}{B} - 1\right)\right) = \frac{L\delta}{2}n\tau, \tag{20}$$

where we define $\delta = \delta_{t,t}$ under Definition 5.1 (maximum step size over the window $\{0, 1, ..., t\}$), so we have $\delta \geq \|\mathbf{x}_t - \mathbf{x}_{t-1}\|$.

Substituting $\nabla f(\mathbf{x}_{t-1}) = \hat{\mathbf{g}}_{t-1} + \mathbf{c}_{t-1}$ into Eq. 17:

$$
\begin{aligned}
f(\mathbf{x}_t) - f(\mathbf{x}_{t-1}) &\leq \hat{\mathbf{g}}_{t-1}^\top (\mathbf{x}_t - \mathbf{x}_{t-1}) + \mathbf{c}_{t-1}^\top (\mathbf{x}_t - \mathbf{x}_{t-1}) + \frac{L}{2}\|\mathbf{x}_t - \mathbf{x}_{t-1}\|^2 \\
&= -\frac{1}{\alpha}\|\mathbf{x}_t - \mathbf{x}_{t-1}\|^2 + \mathbf{c}_{t-1}^\top (\mathbf{x}_t - \mathbf{x}_{t-1}) + \frac{L}{2}\|\mathbf{x}_t - \mathbf{x}_{t-1}\|^2 \\
&\leq \left(-\frac{1}{\alpha} + \frac{L}{2}\right)\|\mathbf{x}_t - \mathbf{x}_{t-1}\|^2 + \|\mathbf{c}_{t-1}\|\|\mathbf{x}_t - \mathbf{x}_{t-1}\| \\
&\leq \left(-\frac{1}{\alpha} + \frac{L}{2}\right)\delta^2 + \left(\frac{L}{2}n\tau\delta\right)\delta \\
&= -\left(\frac{1}{\alpha} - \frac{L}{2}(1 + n\tau)\right)\delta^2 = -C_1\delta^2.
\end{aligned}
\tag{21}
$$

Rearranging terms gives a bound on the step size squared:

$$\delta^2 \leq \frac{f(\mathbf{x}_{t-1}) - f(\mathbf{x}_t)}{C_1}. \tag{22}$$

Next, we upper bound the squared norm of the true gradient. Using Eq. 19 and the bounds established above:

$$
\begin{aligned}
\|\nabla f(\mathbf{x}_{t-1})\|^2 &\leq \|\hat{\mathbf{g}}_{t-1}\|^2 + 2\hat{\mathbf{g}}_{t-1}^\top \mathbf{c}_{t-1} + \|\mathbf{c}_{t-1}\|^2 \\
&\leq \|\hat{\mathbf{g}}_{t-1}\|^2 + 2\|\hat{\mathbf{g}}_{t-1}\|\|\mathbf{c}_{t-1}\| + \|\mathbf{c}_{t-1}\|^2 \\
&\leq \left(\frac{1}{\alpha^2} + \frac{Ln\tau}{\alpha} + \frac{L^2 n^2 \tau^2}{4}\right)\delta^2 = C_2\delta^2.
\end{aligned}
\tag{23}
$$

We now combine these results with the Polyak-Łojasiewicz (PL) condition (Assumption 5.8):

$$2\mu(f(\mathbf{x}_{t-1}) - f(\mathbf{x}^*)) \leq \|\nabla f(\mathbf{x}_{t-1})\|^2 \leq C_2\delta^2 \leq \frac{C_2}{C_1}(f(\mathbf{x}_{t-1}) - f(\mathbf{x}_t)). \tag{24}$$

Rearranging the inequality yields:

$$f(\mathbf{x}_t) - f(\mathbf{x}^*) \leq \left(1 - \frac{2\mu C_1}{C_2}\right)(f(\mathbf{x}_{t-1}) - f(\mathbf{x}^*)). \tag{25}$$

Applying this recurrence $t$ times completes the proof. $\qquad\square$

*Remark* C.2 (A Note on Practical Convergence). Convergence can be much faster in practice than what the theorem suggests, because the constants $C_1$ and $C_2$ are conservative estimates of what they are in practice. The second inequality in Eq. 21 and the second inequality in Eq. 23 overestimate the negative impact of the correction term by over-relaxing the cross terms $\mathbf{c}_{t-1}^\top(\mathbf{x}_t - \mathbf{x}_{t-1})$ and $\mathbf{c}_{t-1}^\top\hat{\mathbf{g}}_{t-1}$ to their worst cases. Therefore, the rate in practice should be smaller than $1 - \frac{2\mu C_1}{C_2}$. Also, the step size $\alpha$ could be set to some values larger than the upper bound discussed in § 5.3 and still let the algorithm stably converge in practice.

**Theorem C.3** (Gradient Estimate Convergence). *Under Assumption 5.7, if the learning rate $\alpha$ is chosen such that $C_1 > 0$ (as defined in Theorem 5.10), the sequence of gradient estimates satisfies*

$$\|\hat{\mathbf{g}}_t - \hat{\mathbf{g}}_{t-1}\| \to 0 \quad as\ t \to \infty.$$

*Proof.* The change in the gradient estimate at step $t$ is given by the difference in partial derivatives for the updated coordinates. Let $I_t$ be the set of $B$ coordinates updated at step $t$. The estimate update rule implies:

$$\hat{\mathbf{g}}_t - \hat{\mathbf{g}}_{t-1} = \sum_{i \in I_t} \left(\nabla_i^\epsilon f(\mathbf{x}_t) - \nabla_i^\epsilon f(\mathbf{x}_{t-\tau-1})\right)\mathbf{e}_i. \tag{26}$$

Using the Lipschitz continuity of the gradient and the fact that the indices in $I_t$ are disjoint:

$$\begin{aligned}
\|\hat{\mathbf{g}}_t - \hat{\mathbf{g}}_{t-1}\| &\leq \sum_{i \in I_t} \|\nabla_i^\epsilon f(\mathbf{x}_t) - \nabla_i^\epsilon f(\mathbf{x}_{t-\tau-1})\| \\
&\leq \sum_{i \in I_t} L_\epsilon \|\mathbf{x}_t - \mathbf{x}_{t-\tau-1}\| \leq \sum_{i \in I_t} L\|\mathbf{x}_t - \mathbf{x}_{t-\tau-1}\| \\
&= BL\|\mathbf{x}_t - \mathbf{x}_{t-\tau-1}\|.
\end{aligned} \tag{27}$$

The distance between inputs over the staleness window $\tau$ can be bounded by the sum of step sizes. Let $\delta = \delta_{t,\tau+1}$ as defined in Definition 5.1. Then:

$$\|\mathbf{x}_t - \mathbf{x}_{t-\tau-1}\| \leq \sum_{k=0}^{\tau} \|\mathbf{x}_{t-k} - \mathbf{x}_{t-k-1}\| \leq (\tau+1)\delta. \tag{28}$$

Substituting this back and noting that $(\tau+1)B = \frac{n}{B} \cdot B = n$:

$$\|\hat{\mathbf{g}}_t - \hat{\mathbf{g}}_{t-1}\| \leq BL(\tau+1)\delta = nL\delta. \tag{29}$$

From the proof of Theorem 5.10 (Eq. 21), we established the sufficient decrease condition:

$$f(\mathbf{x}_{k-1}) - f(\mathbf{x}_k) \geq C_1\|\mathbf{x}_k - \mathbf{x}_{k-1}\|^2. \tag{30}$$

Summing this inequality from $k = 1$ to $t$:

$$f(\mathbf{x}_0) - f(\mathbf{x}_t) \geq C_1 \sum_{k=1}^{t} \|\mathbf{x}_k - \mathbf{x}_{k-1}\|^2. \tag{31}$$

Since $f$ is bounded below, the series on the RHS converges as $t \to \infty$. This implies that the term $\|\mathbf{x}_k - \mathbf{x}_{k-1}\| \to 0$. Consequently, the window maximum $\delta$ also converges to 0.

Combining this with Eq. 29, we conclude:

$$\lim_{t\to\infty} \|\hat{\mathbf{g}}_t - \hat{\mathbf{g}}_{t-1}\| \leq \lim_{t\to\infty} nL\delta = 0. \tag{32}$$

$$\square$$

## D. Pseudocode of CoCD

We provide the pseudocode for Coherent Coordinate Descent (CoCD) below. The core innovation lies in the efficient management of the gradient buffer as a circular FIFO queue, allowing the algorithm to operate on complex, multi-dimensional parameter structures as if they were a contiguous flat vector.

Algorithm 1 details the main optimization step, while Algorithm 2 details the coordinate-wise gradient estimation using the virtualized flattening logic.

---

*Algorithm 1.* Coherent Coordinate Descent (CoCD) - Optimizer Step

---

1: **Input:** Parameters $\Theta = \{\theta_1, \theta_2, \dots\}$, Loss function $\mathcal{L}$, Learning rate $\alpha$, Momentum $\gamma$
2: **Hyperparameters:** Compute Budget $B$, Memory Budget $m$, Perturbation $\epsilon$
3: **State (Persistent):**

- Global Gradient Buffer $\mathcal{G} \in \mathbb{R}^m$ (initialized to 0)

- Pointers: `cur_param_idx`, `cur_weight_idx`, `cur_grad_idx` (initialized to 0)

- Descent Offsets: `grad_offset`, `param_offset`, `weight_offset` (initialized to 0)

4: **Function** STEP($\Theta$):
5:     // 1. Decay existing estimates (Momentum)
6:     $\mathcal{G} \leftarrow \gamma \cdot \mathcal{G}$
7:     // 2. Refresh B coordinates in the circular buffer
8:     APPROXIMATEGRADIENT($\Theta, \mathcal{G}, B, \epsilon$)
9:     // 3. Apply descent using the available gradients in buffer
10:     OPTIMIZE($\Theta, \mathcal{G}, \alpha$)
11:     **return** $\Theta$

---

*Algorithm 2.* Efficient Gradient Estimation & Virtual Flattening

---

1: **Function** APPROXIMATEGRADIENT($\Theta, \mathcal{G}, B, \epsilon$):
2: **for** $k = 1$ **to** $B$ **do**
3:     // Map pointers to specific parameter tensor
4:     $\mathbf{W} \leftarrow \Theta[\texttt{cur\_param\_idx}]$
5:     $w \leftarrow \texttt{cur\_weight\_idx}$
6:     // Zeroth-Order Finite Difference
7:     $v_{old} \leftarrow \mathbf{W}[w]$
8:     $\mathbf{W}[w] \leftarrow v_{old} + \epsilon$
9:     $L_+ \leftarrow \mathcal{L}(\Theta)$
10:     $\mathbf{W}[w] \leftarrow v_{old} - \epsilon$
11:     $L_- \leftarrow \mathcal{L}(\Theta)$
12:     $\mathbf{W}[w] \leftarrow v_{old}$    // Restore parameter
13:     // Store in circular buffer
14:     $\mathcal{G}[\texttt{cur\_grad\_idx}] \leftarrow (L_+ - L_-)/(2\epsilon)$
15:     UPDATEPOINTERS()
16: **end for**
17: **Function** UPDATEPOINTERS():
18: $\texttt{cur\_grad\_idx} \leftarrow (\texttt{cur\_grad\_idx} + 1) \ (\text{mod } M)$
19: $\texttt{cur\_weight\_idx} \leftarrow \texttt{cur\_weight\_idx} + 1$
20: // Check if we finished the current parameter tensor
21: **if** $\texttt{cur\_weight\_idx} \geq \text{numel}(\Theta[\texttt{cur\_param\_idx}])$ **then**
22:     $\texttt{cur\_weight\_idx} \leftarrow 0$
23:     $\texttt{cur\_param\_idx} \leftarrow (\texttt{cur\_param\_idx} + 1) \ (\text{mod } |\Theta|)$
24: **end if**
25: **Function** OPTIMIZE($\Theta, \mathcal{G}, \alpha$):

---

26:  $count \leftarrow 0$
27:  $g\_ptr \leftarrow$ grad_offset
28:  $p\_ptr \leftarrow$ param_offset
29:  $w\_ptr \leftarrow$ weight_offset
30:  **while** $count < m$ **do**
31:      $\mathbf{W} \leftarrow \Theta[p\_ptr]$
32:      $len \leftarrow \text{numel}(\mathbf{W}) - w\_ptr$
33:      $chunk \leftarrow \min(len, m - count)$
34:      // Apply gradients from buffer to parameters
35:      // (Implementation uses flattened views to avoid copies)
36:      $\mathbf{W}[w\_ptr : w\_ptr + chunk] \leftarrow \mathbf{W}[w\_ptr : w\_ptr + chunk] - \alpha \cdot \mathcal{G}[g\_ptr : g\_ptr + chunk]$
37:      // Advance local pointers
38:      $g\_ptr \leftarrow (g\_ptr + chunk) \pmod{m}$
39:      $w\_ptr \leftarrow w\_ptr + chunk$
40:      $count \leftarrow count + chunk$
41:      **if** $w\_ptr \geq \text{numel}(\mathbf{W})$ **then**
42:          $w\_ptr \leftarrow 0$
43:          $p\_ptr \leftarrow (p\_ptr + 1) \pmod{|\Theta|}$
44:      **end if**
45: **end while**

## E. Additional Evaluation: Scaling CoCD to Resnet-20 training

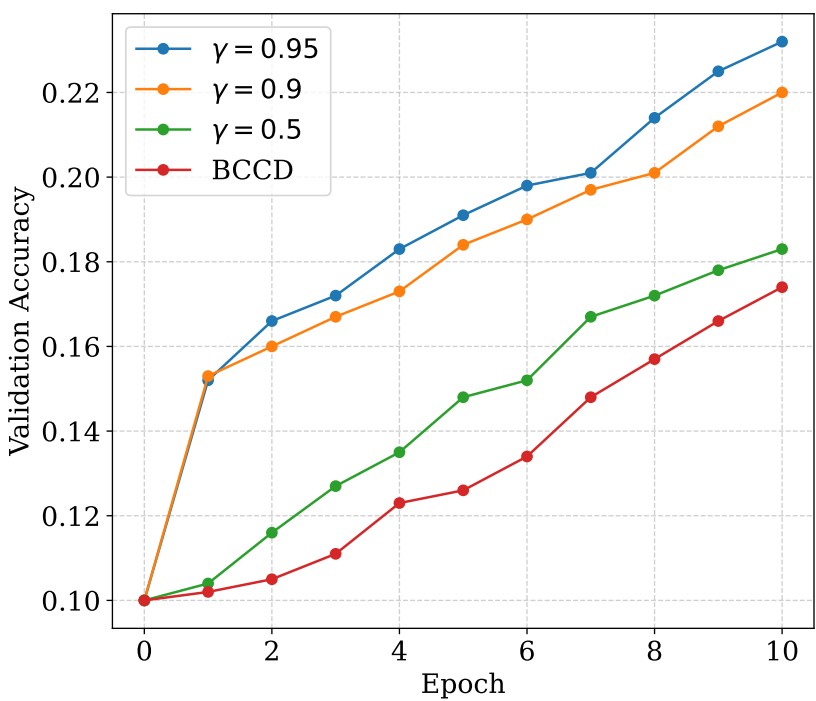

*Figure 6.* Comparison among CoCD with different $\gamma$ values and BCCD on training a Resnet-20 for CIFAR-10 classification.

We train a standard ResNet-20 ($\approx 270k$ parameters) on CIFAR-10. For all settings, we used a learning rate of $0.1$, batch size of $128$, weight decay of $10^{-4}$, budget $B = 2048$, and $\epsilon = 1.0$.

As shown in Fig. 6, within the 10-episode time limit, CoCD with $\gamma \in \{0.5, 0.9, 0.95\}$ outperforms the BCCD baseline. It successfully optimizes the network without staleness leading to instability even when $\gamma$ is high.

