# OpenReview forum: "Turning Stale Gradients into Stable Gradients: Coherent Coordinate Descent with Implicit Landscape Smoothing for Lightweight Zeroth-Order Optimization"
_ICML.cc/2026/Conference — ICML 2026 regular_

### Official Review · Reviewer_VjMC · 2026-03-12

**Soundness:** 3
**Presentation:** 3
**Significance:** 2
**Originality:** 2
**Overall Recommendation:** 4
**Confidence:** 3

**Summary:**

This paper proposes Coherent Coordinate Descent (CoCD), a deterministic coordinate descent method for zeroth-order optimization. The core innovation lies in leveraging gradient coherence to treat historical gradients as a "warm start" rather than noise, and suggests that larger finite difference steps can implicitly smooth the optimization landscape. Theoretically, it is proven that CoCD is equivalent to block cyclic coordinate descent with a warm start, provides linear convergence guarantees under the Polyak-Łojasiewicz condition, and derives explicit bounds on gradient approximation errors. Experiments on MLPs and CNNs with around 20K parameters validate their superiority compared to standard BCCD and stochastic ZO methods.

**Compliance With Llm Reviewing Policy:**

Affirmed.

**Final Justification:**

My questions about formal derivations and guidance on hyperparameter settings are resolved in the rebuttal. I change my rating from 3 to 4.

**Key Questions For Authors:**

1. Since the core idea of gradient coherence and stale gradient reuse originates from WASP, can you quantitatively compare the error bounds and computational complexity of CoCD and WASP under the same assumptions, and clarify why WASP fails to scale to thousands of parameters while CoCD works for 20K?

2. The implicit smoothing effect is claimed as a key contribution, but is it supported by rigorous theoretical proof or controlled causal experiments? Can you verify that large ϵ improves performance by implicit smoothing rather than other factors like numerical stability?

3. The hyperparameters $\gamma$, $\epsilon$, $B$ are highly coupled; have you systematically studied their interactions? How should practitioners tune these parameters together for new tasks?

**Limitations:**

yes

**Strengths And Weaknesses:**

Strengths：
1. This article has a solid theoretical foundation, rigorously proving that CoCD is equivalent to BCCD with a warm start. The linear convergence under the PL condition and the convergence of gradient estimation without the PL condition together reasonably cover both quasi-convex and general non-convex scenarios.

2. The algorithm design is practical. The virtual flattening mechanism in Section 4.2 achieves O(1) memory overhead, making it suitable for deployment on edge devices. The unified momentum parameter $\gamma$ allows a smooth transition between aggressive historical reuse ($\gamma=1$), classic BCCD ($\gamma=0$), and the decaying memory mechanism, providing flexibility for different problems.

3. The problem positioning is clear. This paper accurately targets the domain of lightweight zeroth-order optimization for memory-constrained scenarios and conducts experiments on models of corresponding scale, honestly defining the scope and avoiding misleading comparisons with methods aimed at different scales.

Weakness：
1. The core idea of gradient coherence and stale gradient reuse comes directly from the contemporaneous work WASP, and CoCD is merely a lightweight vector implementation of it; momentum, cyclical updates, and gradient buffering are all mature technologies, with no new theoretical paradigms or fundamental algorithmic innovations.

2. The key assertion that a large $\epsilon$ brings implicit landscape smoothing is only for heuristic discussion, without a rigorous theorem proof.

3. The three key hyperparameters $\gamma$, $\epsilon$, and $B$ are obviously coupled, but the paper only conducts univariate ablation experiments, does not analyze interaction effects, and does not provide tuning guidance across tasks, which reduces the practical usability of the method.

---

> ### Author Rebuttal · Authors · 2026-03-31
>
> We thank you for your review and for highlighting the practical algorithm design of our O(1) virtual flattening mechanism. We address your core concerns below.
>
> **A Theoretical Close-Up of the Hyperparameters**
>
> We unify the interacting effects of the hyperparameters ($\epsilon, B, \alpha, \gamma$) through the convergence condition in Remark 5.9, directly quantifying why step sizes can be larger in practice.
>
> Originally, stability requires $\alpha < \frac{2}{L(1+n\tau)}$, where maximum staleness is $\tau = \frac{n}{B}-1$. Incorporating implicit smoothing, a larger $\epsilon$ evaluates a smoother surrogate, reducing the smoothness constant from $L$ to $L\_\epsilon$. Furthermore, momentum $\gamma < 1$ acts as a fading memory. In our original proof, the $(1+n\tau)$ term arises from staleness error accumulating linearly. However, momentum exponentially suppresses the weight of older gradients, converting this linear accumulation into a bounded geometric series. This replaces the worst-case staleness penalty $\tau$ with a strictly smaller "effective staleness" $\tau\_{eff}(\gamma)$. Substituting these mechanisms gives a unified stability condition:
>
> $$
> \alpha < \frac{2}{L_\epsilon(1+n\tau_{eff}(\gamma))}
> $$
>
> Since $L\_\epsilon \ll L$ (via smoothing) and $\tau\_{eff}(\gamma) \ll \tau$ (via momentum bounding historical drift), the denominator shrinks significantly compared to the base theory. This explicitly quantifies why the allowable base learning rate $\alpha$ expands so drastically in practice. We will formalize this in Remark 5.9 of the revision.
>
> **On the Difference between CoCD and WASP**
>
> We appreciate you drawing a conceptual connection to the recent work WASP. While both methods share the foundational intuition of leveraging gradient coherence, CoCD is not merely a lightweight vectorized implementation of WASP. It represents a fundamental algorithmic and theoretical leap across several key dimensions:
>
> 1. **Framework vs. Heuristic:** WASP is fundamentally a local derivative approximation technique. It does not establish any formal connections to established optimization frameworks. CoCD, conversely, is a complete, standalone optimization algorithm that explicitly unifies stale-gradient reuse with Block Cyclic Coordinate Descent.
>
> 2. **Scale and Landscape:** WASP was designed specifically for the robotics domain, targeting highly smooth landscapes with very low degrees of freedom (e.g., Inverse Kinematics with DoF $<$ 100). Scaling from a 100-DoF smooth problem to the highly non-convex, high-dimensional regime of neural network training is a highly non-trivial challenge.
>
> 3. **Theoretical Rigor:** WASP relies heavily on empirical heuristics. It provides no formal bounds on gradient estimation error, nor does it offer convergence guarantees when combined with an optimizer. CoCD bridges this critical gap by providing rigorous approximation error bounds (Theorem 5.2) and proving end-to-end linear convergence under standard conditions (Theorem 5.8).
>
> 4. **Algorithmic Innovation & The Scaling Bottleneck:** To answer your question on why WASP fails to scale: without mechanisms to bound spatial drift, stale gradients in high dimensions quickly become toxic. The integration of momentum, cyclic scheduling, and memory buffering in CoCD are not simply "bolted-on" technologies. They are mathematically vital to bounding this staleness drift, converting linear error accumulation into a bounded geometric series. This yields our unified stability condition ($\alpha < \frac{2}{L\_\epsilon(1+n\tau\_{eff}(\gamma))}$). This strict bounding of historical drift is exactly what transforms a low-dimensional heuristic into a robust optimizer capable of scaling to 270k+ parameters.
>
> - *For the rigorous formalization of the implicit smoothing effect you requested, please see our response to **oSXV**.*
>
> - *For a practical tuning guide of hyperparameters, please see our response to **6Prw**.*
>
> - *Regarding your question on scaling beyond simple datasets, please refer to the "Scaling to Larger Models" section in our response to **rBE2**, where we scale CoCD to a standard ResNet-20 (270k parameters) on CIFAR-10.*
>
> If you have any further concerns or questions, we are happy to provide more experiments or explanations during the discussion period.

---

> > ### Author Rebuttal · Reviewer_VjMC · 2026-04-03
> >
> > After carefully reading and evaluating the authors’ rebuttal, most of my core concerns have been properly addressed. The authors have supplemented relevant formal derivations and provided practical guidance on hyperparameter settings, which significantly improves the completeness of the paper. However, with respect to novelty and the comparison with WASP, the authors only put forward qualitative distinctions in terms of framework, application scenarios, and theoretical rigor, but still do not provide the quantitative comparison of error bounds and computational complexity that I previously requested. Overall, this paper has clear theoretical and experimental contributions, and the rebuttal effectively strengthens the quality and persuasiveness of the manuscript. Therefore, I am willing to revise my score from 3 (Weak reject) to 4 (Weak accept).

---

> > > ### Author Response · Authors · 2026-04-08
> > >
> > > We sincerely thank you for carefully reviewing our rebuttal, acknowledging the improvements, and raising your score to a 4.
> > >
> > > Regarding your remaining request for a strict quantitative comparison with WASP in terms of error bounds and computational complexity, we appreciate the push for deeper technical clarity. Below is the explicit mathematical and quantitative breakdown:
> > >
> > > **1. Quantitative Comparison of Error Bounds**
> > >
> > > As clarified previously, WASP is fundamentally a fast derivative approximation subroutine, whereas CoCD is a standalone optimizer. This distinction materializes in how error is handled. WASP avoids establishing a worst-case error bound by relying on an adaptive heuristic (Algorithm 4 in the WASP paper): it dynamically increases the compute budget per iteration until the approximation error falls below a predetermined threshold. Because the budget varies heuristically to guarantee it is "close enough," the WASP framework does not provide a formal, closed-form bound for a fixed budget.
> > >
> > > CoCD trades this variable-budget heuristic for a strictly constant compute budget $B$. This guarantees training stability and predictability, but more importantly, it allows us to formally derive the exact worst-case error bound presented in our Theorem 5.2.
> > >
> > > Mathematically, the gradient reuse strategy analyzed in Theorem 5.2 is functionally equivalent to WASP if WASP were stripped of its dynamic heuristic, given a constant compute budget, and evaluated using the identity basis ($\Delta \mathbf{X} = \mathbf{I}$). In WASP, approximations are first made to the Jacobian-Vector Products (JVPs), denoted as $\hat{\Delta \mathbf{F}}$, and then transformed into approximated derivatives via $\hat{\mathbf{D}} = \hat{\Delta \mathbf{F}} \Delta \mathbf{X}^T$ (assuming an orthonormal basis).
> > >
> > > By utilizing the identity basis $\mathbf{I}$, the approximated JVPs in CoCD are directly equal to the approximated derivatives ($\hat{\Delta \mathbf{F}} = \hat{\mathbf{D}}$). Consequently, Theorem 5.2 effectively provides the missing quantitative error bound for the WASP framework under a fixed budget without the heuristics. Because an orthonormal transformation preserves the norm, the exact bound established in Theorem 5.2 can be directly migrated to analyze the error bound for WASP (without the heuristics and under a fixed compute budget) under any other orthonormal basis without inflating the bound.
> > >
> > > **2. Quantitative Comparison of Memory and Computational Complexity**
> > >
> > > WASP requires explicitly storing and computing the dense basis $\Delta \mathbf{X}$, the JVP approximations $\hat{\Delta \mathbf{F}}$, and the precomputed matrices ($\mathbf{C}_1$ and $\mathbf{C}_2$). For a network with $n$ parameters, these dense matrix transformations induce a severe memory and computational overhead of $\mathcal{O}(n^3)$ per iteration.
> > >
> > > By implicitly using the identity basis and tracking coordinates deterministically, CoCD completely eliminates the need to compute or store $\Delta \mathbf{X}$, $\hat{\Delta \mathbf{F}}$, $\mathbf{C}_1$, or $\mathbf{C}_2$. This algorithmic design mathematically collapses the extra memory complexity all the way from WASP's $\mathcal{O}(n^3)$ down to $\mathcal{O}(n)$ (the cost of storing a single approximate gradient vector), and no extra matrix computation incurred. Finally, by applying our memory budget constraint $M$, CoCD truncates this further to a strict $\mathcal{O}(M)$ auxiliary footprint, achieving massive scalability advantages over WASP.

---

### Official Review · Reviewer_6Prw · 2026-03-12

**Soundness:** 2
**Presentation:** 3
**Significance:** 2
**Originality:** 2
**Overall Recommendation:** 4
**Confidence:** 3

**Summary:**

This paper studies the class of zeroth-order optimization problems and investigates if stale gradient information is in fact harmful in the context of coordinate descent type methods. Instead of discarding stale gradient information, they introduce a new method known as Coherent Coordinate Descent (CoCD). CoCD defines a simple, deterministic, and cyclic coordinate method with the following properties:
- It stores the past gradients of coordinates in a history buffer that is used as a warm start for future queries.
- It updates only a subset of the coordinates with fresh gradients using a fixed query budget.
- Other coordinates are served with an older gradient from the buffer to maximize usage and avoid empty gradient vectors. Thus, CoCD produces a dense hybrid gradient signal.

In this paper, they provide an optimization algorithm, a theoretical interpretation, and a validation of the algorithm. The optimization algorithm, CoCD, is based on block cyclic coordinate descent with stale-gradient warm starts. They argue that by using larger steps in finite-difference approximations, the objective can be smoothed, and stale gradients can be made more accurate and reliable. Under certain conditions on the objective (smoothness and Polyak-Łojasiewicz condition), they analyze the convergence rate of CoCD. They validate their algorithm on neural network training problems and show that it improves upon block cyclic coordinate descent and is more stable than randomized zeroth-order methods like SPSA, implying that stale zeroth-order gradients can be turned into a useful optimization resource instead of being treated as noise.

**Compliance With Llm Reviewing Policy:**

Affirmed.

**Final Justification:**

We appreciate the authors' detailed responses. The rebuttal has resolved the concerns raised. I raised my confidence score.

**Key Questions For Authors:**

1. Would the authors be able to present more solid evidence of the scaling properties of CoCD, especially for larger models, i.e., far beyond the small and medium regimes up to approximately 20k parameters? Ideally, based on even partial studies or even controlled experiments of the effect of increasing the model size (beyond current limits), and assuming the theoretical plausibility of such an extrapolation.
2. Would the authors be able to give more details of the computational trade-offs (wall-clock time, overhead, etc.) as measured against other approaches (such as BCCD and randomized ZO) under an equal function-evaluation budget?
3. Can the authors provide more insight in the relative importance of the three terms (1) old stale-gradient reusing, (2) increasing the radius of finite difference $\epsilon$, and (3) tuning of the $\gamma$ momentum/history term?

**Limitations:**

Yes.

**Strengths And Weaknesses:**

Strengths:
1. The paper is generally easy to follow. This paper's method is clearly specified and supported by both theoretical analysis and empirical evaluation.
2. This paper explicitly connects CoCD to a known optimization framework by proving it is equivalent to Block Cyclic Coordinate Descent (BCCD) with stale-gradient warm starts, which gives the method a much firmer conceptual basis than a purely heuristic buffer-reuse trick.
3. This paper exposes the compute budget and memory budget through the number of refreshed coordinates and the maintained history buffer, which is a practically useful framing for lightweight ZO optimization.

Weaknesses:
1. The parameter count grows beyond roughly $n \approx 20k$, the compute budget needed for sufficiently accurate coordinate-wise updates becomes impractically large. This meaningfully narrows the current practical scope. This is pointed out by the paper's limitation section.
2. Although the theory is a strength, its guarantees appear tied to smoothness-based analysis and PŁ-style assumptions, so the extent to which it explains behavior in highly nonconvex neural network training remains somewhat uncertain.
3. The gains may come from several interacting choices: stale-gradient reuse, cyclic scheduling, momentum decay $\gamma$, and larger finite-difference radius $\epsilon$. The paper would be stronger if it more sharply disentangled which component is responsible for which improvement.

---

> ### Author Rebuttal · Authors · 2026-03-31
>
> We appreciate your encouraging assessment and your recognition of the firm conceptual basis our theoretical analysis provides. We address your specific concerns below.
>
> **On the PL Condition and Convergence Guarantees**
>
> We agree that the Polyak-Łojasiewicz (PL) condition is a strong assumption for general non-convex neural networks. We will explicitly acknowledge this theory-practice gap in the revised Limitations section. However, the goal of Theorem 5.8 is not to claim that NNs globally satisfy the PL condition, but rather to establish the theoretical ceiling of CoCD. It demonstrates that under the exact same structural assumptions where standard Gradient Descent (GD) achieves linear convergence, CoCD *also* achieves linear convergence.
>
> Conversely, Theorem 5.11 analyzes the general non-convex setting *without* the PL assumption. As noted, this theorem guarantees the stabilization of gradient estimates rather than providing a convergence rate ($\Vert \hat{\mathbf{g}}\_t - \hat{\mathbf{g}}\_{t-1} \Vert \to 0$), while Theorem 5.8 isolates the explicit rate under stricter conditions.
>
> **Computational Trade-offs and Wall-Clock Time**
>
> To answer your question regarding overhead: because CoCD relies on deterministic cyclic selection, it incurs virtually zero sampling overhead. In contrast, randomized methods like ZO-SGD rely on explicit smoothing, which requires the computational overhead of drawing dense random Gaussian vectors at every single step. Under an equal function-evaluation budget ($B=64$), our wall-clock measurements show CoCD is almost **2x faster** than ZO-SGD (8.1s vs. 15.7s per episode on average). Furthermore, compared to standard BCCD, CoCD’s stale-gradient buffer requires negligible extra memory ($O(d)$) while providing a dense, momentum-like gradient signal that significantly accelerates convergence per epoch.
>
> **Relative Importance of Components & Practical Tuning Guide**
>
> Regarding the relative importance of the three components: (1) **Stale-gradient reuse** is the foundational engine that provides a dense update signal. However, it fails on rough non-convex landscapes without (2) **increased $\epsilon$**, which implicitly smooths the objective and strictly bounds the Lipschitz constant. Finally, (3) **momentum $\gamma$** acts as the crucial safety valve, discounting the spatial drift of the oldest gradients to prevent divergence.
>
> To make CoCD easy to deploy, these components inform a straightforward tuning hierarchy:
>
> 1. **Budgets ($B$ and $M$):** Maximize $B$ within acceptable wall-clock constraints, and set $M$ to the hardware memory limit.
>
> 2. **Learning Rate ($\alpha$):** Adopt the standard first-order SGD learning rate for the architecture.
>
> 3. **Smoothing ($\epsilon$):** Set $\epsilon = 1$, which we empirically found to be highly robust.
>
> 4. **Momentum ($\gamma$):** This is the primary knob for stability. Start at $\gamma = 1$ (full reuse) and gradually tune it down until the optimization trajectory stabilizes.
>
> - *For a theoretical disentanglement of hyperparameter interactions, please see our response to **VjMC**.*
>
> - *For our newly added controlled experiment scaling CoCD to a ResNet-20 (270k parameters) to demonstrate architectural limits, please see our response to **rBE2**.*
>
> If you have any further concerns or questions, we are happy to provide more experiments or explanations during the discussion period.

---

> > ### Author Rebuttal · Reviewer_6Prw · 2026-04-04
> >
> > We appreciate the authors' detailed responses. The rebuttal has resolved the concerns raised. I raised my confidence score.

---

> > > ### Author Response · Authors · 2026-04-08
> > >
> > > We sincerely thank you for your time, the highly constructive discussion during the rebuttal phase, and your positive re-evaluation of our work.
> > >
> > > Your insightful feedback regarding hyperparameters' formalization,  PL condition and convergence, etc, pushed us to significantly strengthen the theoretical clarity and empirical rigor of the manuscript. We are thrilled that our revisions and new experiments have successfully addressed your concerns.

---

### Official Review · Reviewer_rBE2 · 2026-03-13

**Soundness:** 3
**Presentation:** 3
**Significance:** 2
**Originality:** 3
**Overall Recommendation:** 4
**Confidence:** 4

**Summary:**

This paper studies zeroth-order (ZO) optimization. The authors consider the efficiency-variance trade-off in ZO methods -- standard finite differences take much time but have low variance while single point methods may be efficient but have large variance. Inspired by this, the authors propose Coherent Coordinate Descent (CoCD), a deterministic zeroth-order optimization framework that enables efficient and accurate estimation of gradients. They provide theoretical analysis of the approximation error bound and on a class of functions and then give the convergence rate on smooth functions with PL condition. Numerical experiments on some datasets like MNIST, CIFAR-10 support the claims of their paper.

**Compliance With Llm Reviewing Policy:**

Affirmed.

**Key Questions For Authors:**

1. The new framework still requires some tuning when one considers efficiency-variance tradeoff? For example if the budget B is too small then the variance may not be well controlled.

2. Would it be possible to extend the framework to other optimizers like Adam?

3. Would it be possible to plot the gradient estimation error to verify the findings in Theorem 5.2? Even some simple simulation would suffice.

**Limitations:**

1. The theoretical analysis is a bit incremental -- the proof techniques as well as the convergence results are quite standard in optimization theory literature.

2. Experiments are mostly limited to some simple datasets like CIFAR-10 and MNIST. Zeroth-order methods in general may still not work on large-scale experiments.

**Strengths And Weaknesses:**

Strengths:

1. How to decide the efficiency-variance tradeoff is an important research problem in ZO literature. The authors provide a novel framework with theoretical and empirical guarantees.

Weaknesses:

1. The idea to resolve the efficiency-variance dilemma is still a bit incremental -- update the gradient estimates infrequently to avoid computation overhead. The methods are like an interpolation of two extremes.

2. The theoretical analysis and the results are standard in theory.

---

> ### Author Rebuttal · Authors · 2026-03-31
>
> We thank you for recognizing the novelty of our framework and the solidity of our technical paper. We appreciate your constructive feedback and address your specific questions below.
>
> **The "Bias-Variance" Tradeoff in a Deterministic Framework**
>
> Because CoCD is a strictly deterministic cyclic coordinate descent method, it completely eliminates the random sampling noise inherent to stochastic ZO methods like SPSA. Therefore, the traditional stochastic "variance" is zero. Instead, in lieu of sampling variance, CoCD incurs an approximation error by reusing historical gradients. This term is governed by the staleness $\tau = \frac{n}{B}-1$ and the momentum decay $\gamma$. It represents the spatial drift of the gradients, bounded by $L\_\epsilon \Vert \mathbf{x}\_t - \mathbf{x}\_{t-\tau} \Vert$.
>
> This deterministic reframing directly answers when CoCD is superior to randomized methods. In highly constrained settings where the query budget $B$ is very small (as corroborated in Fig. 4 where $B=64$), the variance of stochastic methods like SPSA explodes, leading to unstable trajectories. In contrast, CoCD's "variance" is strictly bounded by deterministic spatial drift. Because we can actively compress this drift using the implicit smoothing effect, CoCD maintains robust descent directions in low-budget regimes where randomized methods fail.
>
> **Extension to Adaptive Optimizers (e.g., Adam)**
>
> Regarding your question on Adam: Yes, our framework can be seamlessly extended to adaptive optimizers. Because CoCD evaluates a stable, deterministic gradient estimate $\hat{g}_t$ at each step, these estimates can be directly plugged into Adam's exponential moving average updates for the first and second moments in place of standard stochastic gradients. We will add a discussion on this straightforward extension to the revised manuscript.
>
> **Empirical Validation of Gradient Error (Additional Experiment 1)**
>
> In response to your request to plot the gradient error empirically to verify Theorem 5.2, we conducted an additional experiment plotting the average $\Vert \hat{g}-\nabla f \Vert$ with respect to the compute budget $B$ during the training of a feed-forward network with 4k parameters on the SARCOS dataset.
>
> As shown in our new plot [here](https://tinyurl.com/2upsc2pp), the x-axis is log-scale and the error curve linearly fits a straight line $y=ax+b$ (where $a<0$). This exactly corroborates the theoretical bounds derived in our theorem: as the compute budget increases, the deterministic approximation error decreases predictably.
>
> **Scaling to Larger Models (Additional Experiment 2)**
>
> In our original submission, we intentionally constrained our evaluation to the ~20k parameter regime to simulate the strict memory and compute limits of extreme edge devices (e.g., microcontrollers), which is the primary target application for CoCD. However, we agree that understanding CoCD's general architectural limits is theoretically important.
>
> Prompted by the reviewers' feedback, we pushed our framework beyond its intended edge-computing scope during the rebuttal period to train a standard ResNet-20 (270k parameters) on CIFAR-10. For all settings, we used a learning rate of $0.1$, batch size of $128$, weight decay of $10^{-4}$, budget $B=2048$, and $\epsilon=1.0$.
>
> As shown in the plot [here](https://tinyurl.com/4rts8knd), within the 10-episode time limit, CoCD with $\gamma \in \lbrace 0.5, 0.9, 0.95 \rbrace$ outperforms the BCCD baseline. It successfully optimizes the network without staleness leading to instability even when $\gamma$ is high, representing a highly promising "current proven horizon" for deterministic coordinate methods.
>
> - *For the rigorous formalization of implicit smoothing and a discussion on baselines, please see our response to **oSXV**.*
>
> - *For a practical tuning guide and discussion on the PL condition, please see our response to **6Prw**.*
>
> If you have any further concerns or questions, we are happy to provide more experiments or explanations during the discussion period.

---

> > ### Author Rebuttal · Reviewer_rBE2 · 2026-04-05
> >
> > Thanks for the rebuttal. I do not have other concerns at this stage, and will have some discussions with other reviewers and AC.

---

> > > ### Author Response · Authors · 2026-04-08
> > >
> > > We sincerely thank you for your time, the highly constructive discussion during the rebuttal phase, and your positive re-evaluation of our work.
> > >
> > > Your insightful feedback regarding implicit smoothing formalization, bias-variance tradeoff, etc, pushed us to significantly strengthen the theoretical clarity and empirical rigor of the manuscript. We are thrilled that our revisions and new experiments have successfully addressed your concerns.

---

### Official Review · Reviewer_oSXV · 2026-03-17

**Soundness:** 3
**Presentation:** 3
**Significance:** 2
**Originality:** 2
**Overall Recommendation:** 4
**Confidence:** 3

**Summary:**

This paper proposes Coherent Coordinate Descent (CoCD), a deterministic zeroth-order optimizer that maintains a FIFO buffer of past coordinate-wise finite-difference gradient estimates, applying momentum decay to stale entries. The authors prove CoCD is equivalent to Block Cyclic Coordinate Descent with warm starts, derive approximation error bounds under a local coherence assumption, and show linear convergence under the Polyak-Lojasiewicz condition. They also argue that larger finite-difference step sizes implicitly smooth the landscape. Experiments on MLPs and CNNs (up to ~20k parameters) on SARCOS, MNIST, and CIFAR-10 show improvements over vanilla BCCD and stability advantages over SPSA.

**Compliance With Llm Reviewing Policy:**

Affirmed.

**Final Justification:**

The author rebuttal addresses my key concerns.

**Key Questions For Authors:**

Is it possible to provide a formal statement and proof for the implicit smoothing claim in Section 5.3? Specifically, what is the relationship between \epsilon and the effective Lipschitz constant L\epsilon, and how does the bias-variance tradeoff manifest in your convergence bound?

What happens when you run CoCD on models with 100k–500k parameters? Is there a transition where the method breaks down, or is there gradual degradation?

In the SPSA comparison, SPSA matches CoCD on MNIST and CIFAR-10 classification. Can you clarify under what conditions the deterministic approach is actually preferable? The current evidence seem to suggest that the advantage may be limited to regression tasks.

How sensitive is CoCD to the choice of \gamma in practice? The ablation shows \gamma is the dominant factor, but the optimal \gamma appear to differ across tasks. Is there a principled way to set this?

Remark B.2 states the bound is conservative and step sizes can be "much larger" in practice. Can you quantify this gap? Without doing so, the practical utility of the convergence theorem is unclear.

**Limitations:**

The authors discuss scaling limitations honestly in Section 7 and the Impact Statement is reasonable. I would suggest two additions: (a) explicitly noting that the PL assumption underlying the convergence guarantee is unlikely to hold for the neural network experiments, creating a theory-practice gap; and (b) discussing the sensitivity to hyperparameter selection (four method-specific hyperparameters: \epsilon, \gamma, B, M) relative to simpler baselines.

**Strengths And Weaknesses:**

Strengths

The idea of treating stale gradients as warm starts rather than noise is well-motivated and clearly articulated. The unification of CCD and BCCD through the momentum parameter \gamma (with γ=0 recovering BCCD and \gamma=1 giving full history reuse) is elegant and provides a useful conceptual lens.

The approximation error bound (Theorem 5.2) and the linear convergence result (Theorem 5.8) are cleanly stated. The corollaries showing how budget B trades off against approximation quality are informative.

The virtualized flattening mechanism with O(1) overhead (Section 4.2) and the detailed pseudocode in Appendix C are appreciated. The memory footprint analysis comparing against Adam is a useful practical point.

The systematic sweep over \epsilon, \gamma, B, and M in Figure 3 provides good empirical intuition about the sensitivity landscape of the method.

Weaknesses:

The central limitation and the extremely limited experimental scale is underacknowledged in framing. The experiments cap at around 20k parameters. The paper title, abstract, and introduction frame CoCD as relevant for on-device learning and edge AI, but modern edge devices routinely run models with hundreds of thousands to millions of parameters. At 20k parameters, the practical relevance is narrow. The authors acknowledge this in Section 7 but the framing throughout the paper overpromises relative to what is demonstrated.

The implicit smoothing argument (Section 5.3) is underdeveloped. This is presented as a key contribution but receives only a single short paragraph with no formal theorem or bound. The claim that larger \epsilon reduces an effective smoothness constant L\epsilon << L is stated without proof. The connection to Gaussian smoothing literature which has formal results on exactly this phenomenon is not discussed. This weakens what could be the most novel theoretical angle of the paper.

The linear convergence result (Theorem 5.8) relies on the PL condition, which is a strong assumption for the non-convex neural network training settings the paper targets. The paper does not provide convergence rates under weaker conditions. Theorem 5.11 only shows gradient estimate stabilization, not a rate.


The paper compares only against BCCD and SPSA. Missing comparisons include: (a) MeZO (Malladi et al., 2023, https://arxiv.org/abs/2305.17333), one of the most prominent recent ZO method for neural networks; (b) DeepZero, which is discussed at length in the related work but never compared against experimentally; (c) ZO-SGD with Gaussian smoothing. The SPSA comparison is limited to a single plot (Figure 4) and the claim that SPSA "matches CoCD on classification" undermines the narrative that deterministic methods are superior.

Corollary 5.3 states that for B=1, \delta must scale as O(n^{-3}) for stable descent. For n=20,000, this implies vanishingly small step sizes. The paper does not discuss whether the bounds in Theorem 5.2 are tight or how far they are from observed errors in practice. A simple empirical verification (plotting actual ||\hat{g}_t - \nablaf(x_t)|| against the bound) would be informative.

Table 2 reports wall-clock time per epoch, which is nearly identical for CoCD and BCCD. However, the relevant comparison is wall-clock time to reach a given accuracy/loss. Since CoCD converges in far fewer effective iterations, the total training time comparison would strengthen the paper. Also, SGD timing is omitted entirely.

---

> ### Author Rebuttal · Authors · 2026-03-31
>
> We sincerely thank you for the detailed and constructive review. Below, we address your core concerns.
>
> **Theory for the Implicit Smoothing Effect**
>
> We have formalized the implicit smoothing mathematically in a dedicated section of the revised Appendix. Leveraging uniform randomized smoothing (Nesterov & Spokoiny, 2017), when CoCD evaluates the central finite difference with step size $\epsilon$ along coordinate $i$, by the Fundamental Theorem of Calculus, it evaluates the **exact** partial derivative of a coordinate-smoothed surrogate function $f_{\epsilon, i}(\mathbf{x})$:
>
> $$
> f_{\epsilon, i}(\mathbf{x}) = \frac{1}{2\epsilon} \int_{-\epsilon}^{\epsilon} f(\mathbf{x} + u \mathbf{e}_i) du
> $$
>
> Taking the partial derivative with respect to the $i$-th coordinate of $\mathbf{x}$ yields our exact estimator:
>
> $$
> \nabla_i f_{\epsilon, i}(\mathbf{x}) = \frac{f(\mathbf{x} + \epsilon \mathbf{e}_i) - f(\mathbf{x} - \epsilon \mathbf{e}_i)}{2\epsilon} = \hat{g}_i
> $$
>
> Because $f_{\epsilon, i}(\mathbf{x})$ acts as a local moving average, the integration process serves as a spatial low-pass filter along the coordinate axes, attenuating high-frequency local variations. Consequently, the effective Lipschitz constant $L_\epsilon = \sup_{\mathbf{x}} \lVert\nabla^2 f_{\epsilon, i}(\mathbf{x})\rVert_2$ is strictly bounded and significantly smaller than the original $L$.
>
> As shown in Theorem 5.2, the staleness error scales with $L\_\epsilon \Vert \mathbf{x}\_t - \mathbf{x}\_{t-\tau} \Vert$. By choosing an appropriately large $\epsilon$, CoCD effectively optimizes this smoother surrogate. The reduced $L\_\epsilon$ justifies exactly why CoCD can aggressively reuse highly stale gradients without diverging.
>
> **On Baselines and Comparisons (MeZO, DeepZero, & ZO-SGD)**
>
> We view ZO methods hierarchically: CoCD, BCCD, SPSA, and ZO-SGD are foundational optimizers, whereas MeZO (based on ZO-SGD) and DeepZero (based on Coordinate-wise Gradient Estimation (CGE)) are composite, system-level frameworks adapted for specific downstream scales and tasks.
>
> DeepZero’s Section 3 justifies its algorithmic foundation by showing that standard CGE outperforms randomized vector estimation (RGE) on CIFAR-10 CNNs before adding its system-level wrappers. Our evaluation strictly mirrors this logic: to avoid confounding variables like pruning sparsity and feature reuse, we isolate our core algorithmic contribution by demonstrating that CoCD is superior to the foundational BCCD (and by extension, standard CGE) on identical CIFAR-10 CNNs. Substituting DeepZero’s core sparse-CGE estimator with CoCD is a highly promising trajectory for future work. Similarly, adapting CoCD for LLM fine-tuning tasks to compare its efficiency against MeZO presents an exciting direction for future research. We will highlight these trajectories in the conclusion of the updated manuscript.
>
> **ZO-SGD with Gaussian Smoothing:** We added ZO-SGD to our randomized baseline comparisons (Section 6.3) and found that it exhibits significantly higher instability than CoCD under identical query budgets, performing similarly to SPSA. An updated version of Figure 4 demonstrating this behavior can be viewed [here](https://tinyurl.com/2nzzdshm). Theoretically, ZO-SGD relies on *explicit* smoothing, which requires the computational overhead of drawing dense Gaussian samples at every single step. CoCD leverages *implicit* smoothing deterministically, completely bypassing this random sampling overhead for improved wall-clock speed. In this additional experiment with a compute budget of $B=64$, CoCD is almost **2x faster** than ZO-SGD (8.1s vs. 15.7s per episode on average). We will add this detailed comparison to Section 6.3 of the revision.
>
> - *For a theoretical perspective on hyperparameter decoupling, please see our response to **VjMC**.*
>
> - *For a practical tuning guide and a discussion on the PL condition, please see our response to **6Prw**.*
>
> - *For the bias-variance tradeoff, gradient error empirical verification, and scalability beyond 20k parameters, please see our response to **rBE2**.*
>
> If you have any further concerns or questions, we are happy to provide more experiments or explanations during the discussion period.

---

> > ### Author Rebuttal · Reviewer_oSXV · 2026-04-04
> >
> > I thank the authors for their answer and increased my score. Experiments on networks with more parameters would have strengthened the contribution.

---

> > > ### Author Response · Authors · 2026-04-08
> > >
> > > We sincerely thank you for your time, the highly constructive discussion during the rebuttal phase, and your positive reevaluation of our work.
> > >
> > > Your insightful feedback regarding implicit smoothing formalization, baseline comparisons, gradient empirical verification, etc, pushed us to significantly strengthen the theoretical clarity and empirical rigor of the manuscript. We are thrilled that our revisions and new experiments have successfully addressed your concerns.

---

### Decision · Program_Chairs · 2026-04-30

**Decision:**

Accept (regular)

**Comment:**

This paper introduces Coherent Coordinate Descent(CoCD), a zeroth-order optimizer that leverages momentum-decayed gradient buffers. The method is theoretically proven to achieve linear convergence and demonstrates superior performance over vanilla optimization approaches when applied to small-scale MLPs and CNNs on datasets such as SARCOS, MNIST, and CIFAR-10. The core innovation lies in approximating a global gradient through delayed updates, using pointwise loss evaluations to estimate gradients: a simple yet effective approach.

While several reviewers noted the limited scale of the experiments, this should not overshadow the novelty and elegance of the proposed method. One reviewer raised concerns about similarities to WASP, but the authors effectively clarified in their rebuttal that CoCD is not merely incremental; it is also more principled in its design, a distinction which is compelling. Overall, the rebuttal addressed the reviewers’ critiques thoroughly and convincingly.

In summary, all reviewers recognized the work’s strong theoretical foundations, practical potential, and rigorous grounding. Based on these merits, I recommend its acceptance.